# Single-cell mapping of lipid metabolites using an infrared probe in human-derived model systems

Yeran Bai [1,2] ✉, Carolina M. Camargo[1], Stella M. K. Glasauer[1], Raymond Gifford[1], Xinran Tian[1], Andrew P. Longhini[1] & Kenneth S. Kosik [1] ✉

Understanding metabolic heterogeneity is the key to uncovering the underlying mechanisms of metabolic-related diseases. Current metabolic imaging studies suffer from limitations including low resolution and specificity, and the model systems utilized often lack human relevance. Here, we present a single-cell metabolic imaging platform to enable direct imaging of lipid metabolism with high specificity in various human-derived 2D and 3D culture systems. Through the incorporation of an azide-tagged infrared probe, selective detection of newly synthesized lipids in cells and tissue became possible, while simultaneous fluorescence imaging enabled cell-type identification in complex tissues. In proof-of-concept experiments, newly synthesized lipids were directly visualized in human-relevant model systems among different cell types, mutation status, differentiation stages, and over time. We identified upregulated lipid metabolism in progranulin-knockdown human induced pluripotent stem cells and in their differentiated microglia cells. Furthermore, we observed that neurons in brain organoids exhibited a significantly lower lipid metabolism compared to astrocytes.

Metabolic heterogeneity is prevalent in biological systems and has profound influences on human health, including diseases such as diabetes, cancers, and neurodegenerative disorders[1–3]. To accurately model human-relevant metabolism and develop effective therapies, it is crucial to use appropriate model systems. Traditional animal models engineered to produce phenotypic features of human diseases like *Caenorhabditis elegans*, *Drosophila melanogaster*, and mice have been foundational in metabolic research[4–8]. They have been extensively characterized, providing a vast pool of resources and a unique environment to study tissue interactions. However, despite their undeniable contributions, these models often exhibit metabolic pathways that significantly diverge from those found in humans[9–11]. On the other hand, human-derived models, such as primary human cells, human induced pluripotent stem cells (hiPSCs), and hiPSC-derived organoids offer distinct advantages. These systems more closely recapitulate the heterogeneity and complexity of human metabolic processes[12–14]. They

also offer the potential for more direct applicability to human physiology, thereby increasing the translational value of research findings[15–17]. To better understand metabolic heterogeneity within these models, advanced metabolic imaging technologies are needed. Positron emission tomography (PET) and magnetic resonance imaging (MRI) have been widely used in clinics[18,19], but their spatial resolution is not sufficient to study these model systems. Imaging mass spectrometry (IMS) provides detailed metabolomic characterization[20,21]. However, most commercially available IMS setups have a spatial resolution of tens of micrometers[21,22], which makes single-cell metabolic analysis difficult to achieve. Optical imaging, on the other hand, provides sub-micrometer spatial resolution, which is an attractive alternative for single-cell metabolic analysis. General metabolic activity levels can be quantified by detecting the intensity or lifetime of fluorescent coenzymes nicotinamide adenine (pyridine) dinucleotide and flavin adenine dinucleotide[23,24]. To investigate certain metabolic

[1]Neuroscience Research Institute, Department of Molecular, Cellular, and Developmental Biology, University of California, Santa Barbara, CA, USA. [2]Photothermal Spectroscopy Corp., Santa Barbara, CA, USA. ✉e-mail: yrbai@ucsb.edu; kosik@lifesci.ucsb.edu

pathways, modified exogenous molecules such as fluorescence analogs of glucose and lipids have also been used[25,26]. Nevertheless, these modified molecules contain fluorescence tags that may perturb normal cell physiology and lead to altered pathways when compared to unmodified counterparts[27,28].

Vibrational spectroscopy imaging integrated with vibrational probes provides a new direction for single-cell metabolic analysis. Vibrational probes are biorthogonal, enabling selective detection of metabolic products without interference from cellular endogenous molecules[8,29–32]. Due to their small size and bio-compatibility, vibrational probe-labeled small molecules are widely used to trace metabolic activities such as fatty acids, amino acids, and nucleic acids. Existing cellular metabolic profiling with vibrational probes is largely focused on Raman imaging, where a full set of Raman probes have been investigated[33,34]. Infrared (IR) absorption provides over eight orders of magnitude stronger absorption cross sections when compared with Raman scattering[35], which promises higher sensitivity and speed. While there are a handful of reports integrating different IR probes to study metabolism using Fourier transform IR or discrete frequency IR[29,36], the coarse resolution associated with these IR imaging modalities presents challenges to achieving single-cell or even sub-cellular metabolic analysis and co-registration of other imaging modalities such as fluorescence imaging.

The recently developed optical photothermal infrared (OPTIR) microscope provides sub-micrometer resolution IR imaging in the far-field and is highly compatible with fluorescence imaging[37–40]. In OPTIR, a visible probe beam is used to detect the localized photothermal effect induced by sample absorption of an IR pump source. Isotope-labeled substrates such as fatty acids or glucose have been used to study the metabolic activity of protein and lipid synthesis with sub-cellular resolution[41–43]. However, carbon-deuterium bonds produce weak IR signals[44], and $^{13}$C labeled molecules suffer from background signals because the shifted peaks are largely overlapped with the unlabeled molecules[43]. Therefore, it is vital to investigate and incorporate sensitive IR probes into OPTIR systems that enable background-free detection of metabolic products.

In this report, we demonstrate OPTIR metabolic imaging integrated with IR probes for single-cell metabolic analysis in human-relevant model systems consisting of neuroglioma, hiPSCs, hiPSC-derived microglia, and hiPSC-derived brain organoids. By using a sensitive and biorthogonal IR tag azide, we were able to selectively image newly synthesized lipids in these model systems with sub-micrometer resolution. We also investigated the differences in lipid metabolism between hiPSC-derived microglia cells and hiPSCs, and the impact of progranulin deficiency on lipid metabolism. To exemplify the potential of this platform for cell-type specific metabolic imaging, we compared the lipid metabolic levels of neurons and astrocytes in hiPSC-derived brain organoids. These demonstrations validate the applicability of the IR probe integrated OPTIR platform for single-cell lipid metabolic imaging and the suitability of using human-derived model systems for studying human-relevant metabolic alterations.

## Results

### Fluorescence-integrated OPTIR platform for lipid metabolic imaging

The concept of utilizing IR probes to trace metabolic processes is illustrated in Fig. 1A. Azide was used as the IR tag since it offers an at least one order of magnitude increase in the absorption cross-section compared with the C-D stretching mode[35] (Supplementary Fig. 1). Additionally, the absorption peak of azide lies in the so-called cell-silent region, which enables selective detection of newly synthesized molecules without interference from endogenous cellular signals. Palmitic acid (PA) is the first fatty acid produced during fatty acid synthesis in mammalian cells and can be further desaturated and elongated to generate an array of fatty acids[45]. Moreover, PA

metabolism has been reported to be dysregulated in various diseases including Alzheimer's disease[46,47]. Therefore, we aim to demonstrate the feasibility of direct imaging of sub-cellular lipid metabolism using PA as a model compound. Azide conjugated PA (azide-PA) was added into the culture media of cells and tissue, and subsequently azide tags were incorporated into the newly-synthesized lipids such as triglycerides, phospholipids, and cholesterol esters through different metabolic processes. By focusing OPTIR imaging at the azide stretching frequency, newly synthesized lipids were mapped with high resolution.

The schematic setup of the fluorescence-integrated OPTIR instrumentation is shown in Fig. 1B. Briefly, a pulsed quantum cascade laser is used as the mid-IR pump source with a wavelength coverage of the fingerprint region (940 to 1800 cm$^{-1}$, or 5.6 to 10.6 μm) and cell-silent region (2000 to 2320 cm$^{-1}$, or 4.3 to 5 μm). The visible probe source is provided by a 532 nm continuous wave laser. The mid-IR and visible beams were focused on the same spot on the sample with a reflective or a refractive objective. For cell imaging, a counter-propagation of the pump and probe beam was used, while a co-propagation was utilized for organoid slice imaging. Backscattered probe photons were collected and guided to a photodiode. Signals were demodulated with a lock-in amplifier at the mid-IR laser repetition rate of 100 kHz. OPTIR images were acquired in a raster-scanning manner with a motorized stage. The fluorescence imaging module was equipped with multiple fluorescence cubes and a high-quantum yield camera.

To demonstrate the biorthogonal capacity of using azide-PA to monitor lipid metabolism, we acquired OPTIR spectra of azide-PA and PA (Fig. 1C). In addition to the major peaks in the fingerprint region including peaks around 1700 cm$^{-1}$ ($\nu_{C=O}$) and 1473 cm$^{-1}$ ($\delta_{C-H}$), azide-PA possesses an additional peak centered around 2100 cm$^{-1}$, which corresponds to -N$_3$ stretching mode. A typical biological cell spectrum is shown in red as a reference, and it is clear the azide peak lies in the region where no cellular endogenous signal is present. We further evaluated the limit of detection (LoD) of azide-PA using spectra from its serial dilution in dimethyl sulfoxide (DMSO) (Supplementary Fig. 2). The LoD of azide bond in DMSO is around 100 μM, which surpasses the LoD of the alkyne bond in SRS microscopy[48]. The LoD is limited by the relatively low NA objective and non-optimized detection geometry. A recent study has achieved a 5 μM LoD for the nitrile bond in OPTIR microscopy using a 1.2NA water-immersion objective and a fast digitization method[49]. Given the higher extinction coefficient of azide compared to nitrile[50,51], we expect an improved LoD with optimized experimental conditions such as co-propagation of IR and visible light, along with transmission detection using a high numerical aperture liquid immersion objective lens with cover glass correction. Such an approach is anticipated to enhance the photon collection efficiency, thereby improving the LoD of target molecules[52].

### Newly synthesized lipids were directly visualized in cells

After characterizing the system with pure samples, we aimed to test the feasibility of utilizing OPTIR and azide-PA to image lipid metabolism in cells. To map azide incorporation into cells, the mid-IR laser was tuned to 2096 cm$^{-1}$ targeting -N$_3$ stretching mode. To visualize the total lipid distribution, we acquired OPTIR images around 1744 cm$^{-1}$ targeting C = O stretching mode in lipid esters. We incubated human neuroglioma H4 cells with media containing azide-PA at a final concentration of 100 μM for around 6.5 h, then fixed them for imaging. This concentration was selected based on prior metabolic studies using vibrational probes and cell viability tests against different PA concentrations[29,53–55]. We also observed the incorporation of azide into intracellular lipids at a reduced azide-PA concentration of 20 μM (Supplementary Fig. 3), suggesting that a lower azide-PA incubation concentration is viable for tracing lipid metabolites, especially if there are viability concerns for certain cell types. We first verified that OPTIR

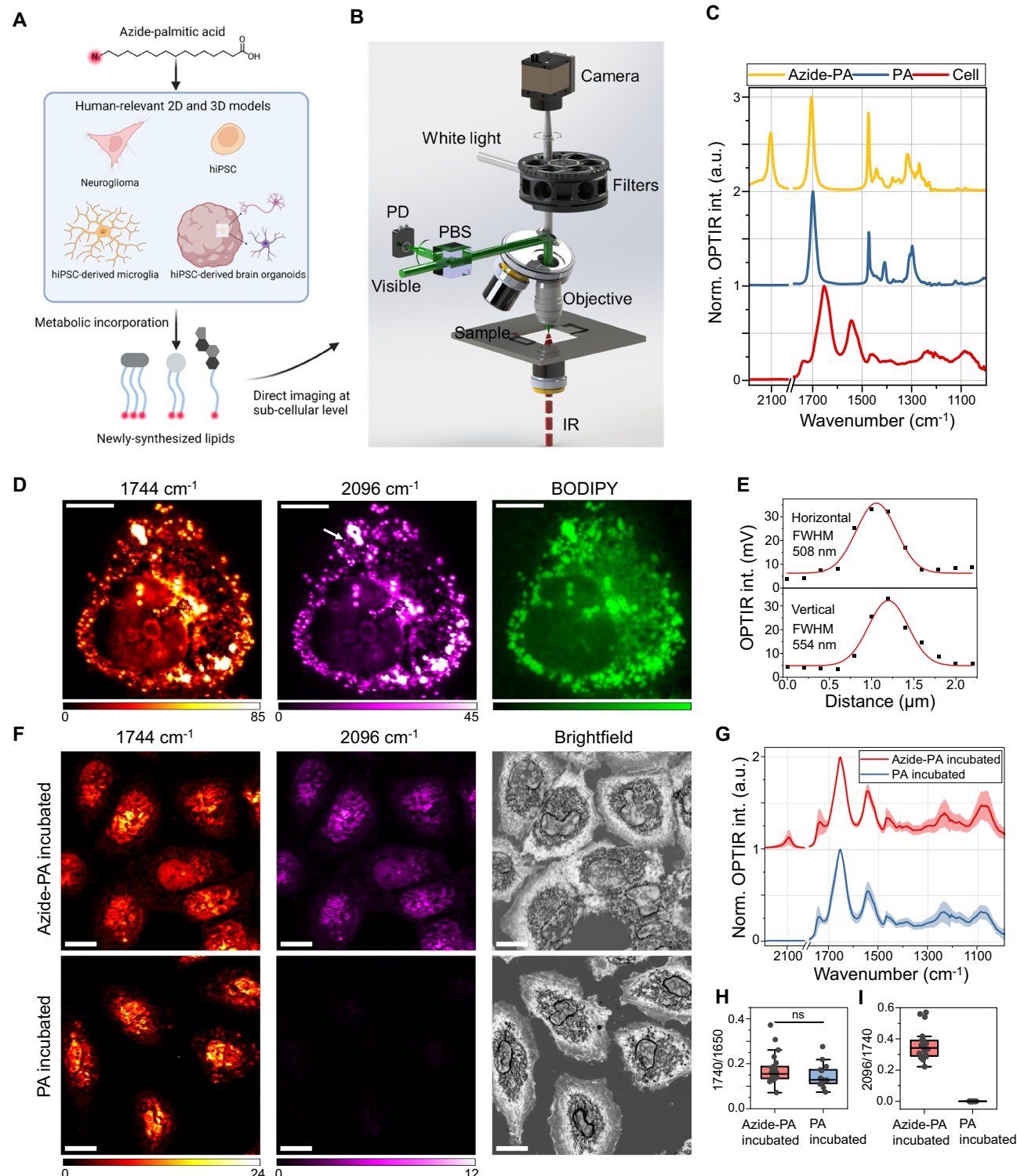

image contrasts at 1744 cm⁻¹ and 2096 cm⁻¹ were indeed from lipids by comparing them with fluorescence images of lipid staining with BODIPY (Fig. 1D). A good agreement between OPTIR and BODIPY images validates that the contrasts were from lipids, further underscoring the chemical selectivity of OPTIR imaging. In addition to the incorporation of azide-tagged lipids into intracellular lipid droplets, we also observed their incorporation into plasma membranes (Supplementary Fig. 4). Since the concentration of lipids is lower in the cell membrane when compared to lipid droplets, the signal contrast is weaker. Despite this, we still managed to observe clear cellular

boundaries. Additionally, the azide peak was unambiguously resolved on the spectrum when localized at the cell boundary.

We further quantified the spatial resolution of OPTIR metabolic imaging by pinging to a small lipid droplet and acquiring a line profile along two, perpendicular directions (Fig. 1E). The fitting results indicated a resolution of around 500 nm, which is a 6 times improvement compared to previously reported IR-based metabolic imaging setups[29]. Another key advantage of OPTIR compared to direct IR imaging is the consistent sub-micrometer spatial resolution irrespective of broadly tuning IR wavelengths (4.3 to 10.6 μm),

**Fig. 1 | Concept, instrumentation, and characterization of imaging lipid metabolism with OPTIR microscopy and IR probes. A** Azide tags were metabolically incorporated into newly-synthesized lipids and thus enabled the selective detection of these metabolites. **B** Schematic setup of a fluorescence-integrated OPTIR system. IR infrared, PBS polarizing beam splitter, PD photodiode. **C** OPTIR spectra of Azide-PA, PA, and a biological cell. Spectra were normalized (Norm.) to maximum intensity and offset for clarity. **D** Representative total lipids (1744 cm$^{-1}$), newly synthesized lipid (2096 cm$^{-1}$), and BODIPY imaging from a single cell. Scale bars, 10 µm. **E** Line profile along the horizontal and vertical direction of a small lipid droplet indicated with a white arrow in 2096 cm$^{-1}$ image in **D**. The raw data is shown in black dots and the Gaussian fitting is shown in the red curve. Full-width half maximum (FWHM) of the fitted curve is shown to demonstrate the spatial resolution. **F** Representative OPTIR images at 1744 cm$^{-1}$ and 2096 cm$^{-1}$ of human neuroglioma H4 cells after incubation in Azide-PA and PA-containing media for 24 h. Corresponding brightfield images are also shown. Scale bars, 20 µm **G** OPTIR spectra for azide-PA and PA incubated cells. Raw spectra were normalized to protein signal at 1654 cm$^{-1}$ and offset for clarity. Mean curve (solid) and standard deviation (shade) were generated from 22 azide-PA cultured cells and 14 PA cultured cells. **H, I** Statistical analysis of total lipid and newly synthesized lipid from spectral fitting results. Area under curve centered around 1654 cm$^{-1}$, 1740 cm$^{-1}$, and 2096 cm$^{-1}$ were used for quantification. A non-significant difference was observed for total lipids between azide-PA incubated ($n = 22$) and PA incubated cells ($n = 14$). Statistical test: two-sided two-sample t-test **H**. Central horizontal lines in the box plot indicate medians, box limits indicate first and third quartiles, vertical whisker lines indicate minimal and maximum values, the outliers were identified using a coefficient of 1.5 times the interquartile range (H-I). The spectral range of 1780 to 2030 cm$^{-1}$ was omitted since no observable peaks were presented. Source data are provided as a Source Data file.

whereas direct IR imaging will experience 2.5 times reduced resolution when IR tuning in this range.

To demonstrate the capability of selectively probing of newly synthesized lipids enabled by the azide probe, we imaged 1744 cm$^{-1}$ and 2096 cm$^{-1}$ channels from H4 cells incubated with azide-PA or PA-containing media at a final concentration of 100 µM for 24 h. Clear contrasts were observed in the 2096 cm$^{-1}$ channel for azide-PA incubated cells, indicating the active incorporation of the azide molecule to newly synthesized lipids. As a control, cells incubated with unmodified PA do not produce the spectroscopic signatures of newly synthesized lipids, thus no contrast was observed at 2096 cm$^{-1}$ (Fig. 1F). We randomly picked points in the cytoplasm from multiple cells in each condition to perform spectral measurements and analyses (Fig. 1G–I). We performed whole-spectral fitting (Supplementary Fig. 5) to decompose the signal from protein (amide I ~ 1650cm$^{-1}$), total lipids (carbonyl ~ 1740 cm$^{-1}$), and newly synthesized lipids (azide ~ 2096 cm$^{-1}$). To account for focus and biomass variations across different fields of view (FOV), we did not rely solely on a single-color imaging or an individual fitted peak for total lipid quantification. Instead, we normalized the total lipid to protein (1740/1650), thereby producing a more reliable normalized total lipid signal. Furthermore, we defined the ratio of newly synthesized lipids to total lipids as 2096 cm$^{-1}$/1740 cm$^{-1}$. The spectral-based quantification showed no significant difference in the normalized total lipid between incubations with PA or azide-PA, indicating the physiological compatibility of azide-PA (Fig. 1H). The newly synthesized lipid levels showed a significant difference between azide-PA and PA treatment, as expected (Fig. 1I).

To ascertain the observed signal is not from fatty acids uptake, but from the newly synthesized lipids after metabolic process, we chemically interrogate the lipid metabolic pathway by treating neuroglioma cells with the small molecule inhibitor Triacsin C. Triacsin C is a long fatty acyl CoA synthetase inhibitor that blocks the de novo synthesis of glycerolipids and cholesterol esters[56]. For the Triacsin C treated group, 1 µM of the inhibitor was added together with azide-PA and cultured for 24 h. For the control group, cells were incubated with azide-PA for 24 h. We acquired images from both groups and the representative images are shown in Supplementary Fig. 6. It was clear that with the Triacsin C treated group, both the 2096 cm$^{-1}$ and 1744 cm$^{-1}$ contrast drops substantially. We further acquired spectra from treated and control cells and observed an obvious reduction of 1740 and 2096 cm$^{-1}$ peak intensity for the Triacsin C treated group (Supplementary Fig. 6B). Imaging and quantification results confirmed the normalized total lipid as well as the ratio of newly synthesized to total lipid both significantly decreased after 24 h treatment of Triacsin C, suggesting the observed signal in 2096 cm$^{-1}$ was indeed from the newly synthesized lipids.

To investigate the dynamics of lipid synthesis following the supplement of azide-PA, we tracked the time-dependent incorporation of azide into newly synthesized lipids over 24 h in culture (initialized azide-PA final concentration 100 µM). The neuroglioma H4 cells were collected at various incubation times and fixed for OPTIR imaging. Representative OPTIR images at 1744 and 2096 cm$^{-1}$ are shown in Fig. 2A. The azide signal was detectable as early as 1 h posttreatment, followed by an increase till -11 h, followed by a gradual decrease. We found a similar contrast trend for the total lipids signal. The spectra and quantification results showed the intensity of normalized total lipids and newly synthesized lipids reached a maximum between 6.5 to 11 h followed by a subsequent decrease (Fig. 2B–D). The ratio of newly synthesized lipids to total lipids follows a similar pattern while staying relatively stable longer (6.5 to 16 h) before finally decreasing (Fig. 2E). The observed rise in total lipids and newly synthesized lipids during the initial hours indicates active lipid synthesis. Since the fatty acid concentration in the cell culture media[57] is relatively low compared to the added azide-PA, cells are more likely to uptake azide-PA, leading to an increased azide signal. The observed decline of total lipids and newly synthesized lipids after 11 to 16 h might be attributed to cellular metabolic adaptions due to decreasing nutrient availability. It has been shown that PA treatment can promote cancer cell growth[58]. With active cellular growth and division, there is an increased demand for lipids, which are essential for membrane expansion and phospholipids synthesis. As exogenous lipid resources are reduced, cells may initiate lipid catabolism to support the increasing lipid demand. This pattern of an initial lipid surge followed by a decline after PA treatment has been described previously in a different cell line[59]. Interestingly, the same research highlighted distinct lipid dynamics upon oleic acids treatment, where it shows a continuous buildup of lipids over 24 h incubation period. These results indicate that specific fatty acids may trigger distinct lipid metabolic pathways, leading to varied lipid dynamics. To thoroughly understand the unique lipid metabolism dynamics that we observed in the present study, it is essential to perform comprehensive analysis across diverse fatty acids and cell types.

## Enhanced lipid metabolism in hiPSC during microglia differentiation

After characterizing lipid metabolism of immortalized cells using OPTIR and azide-PA, we then examined the potential of using this platform to study lipid metabolism in hiPSC and hiPSC-derived microglia. Microglia cells were differentiated from hiPSCs following a published protocol[60]. The hiPSCs and differentiated microglia cells were incubated with azide-PA at a final concentration of 100 µM for 24 h, and then fixed before OPTIR imaging. We validated our microglia differentiation protocol by labeling with the microglia markers IBA1 and CD45 (Supplementary Fig. 7). A representative single-cell image is shown in Fig. 3A. In the hiPSC-derived microglia, both the carbonyl and azide signals are prominent, and there is a substantial overlap between the two contrasts (Supplementary Fig. 8). On the other hand, hiPSCs produced a notably reduced signal in both channels. From the OPTIR spectra (Fig. 3B), microglia showed an overall increase in the 1740 cm$^{-1}$

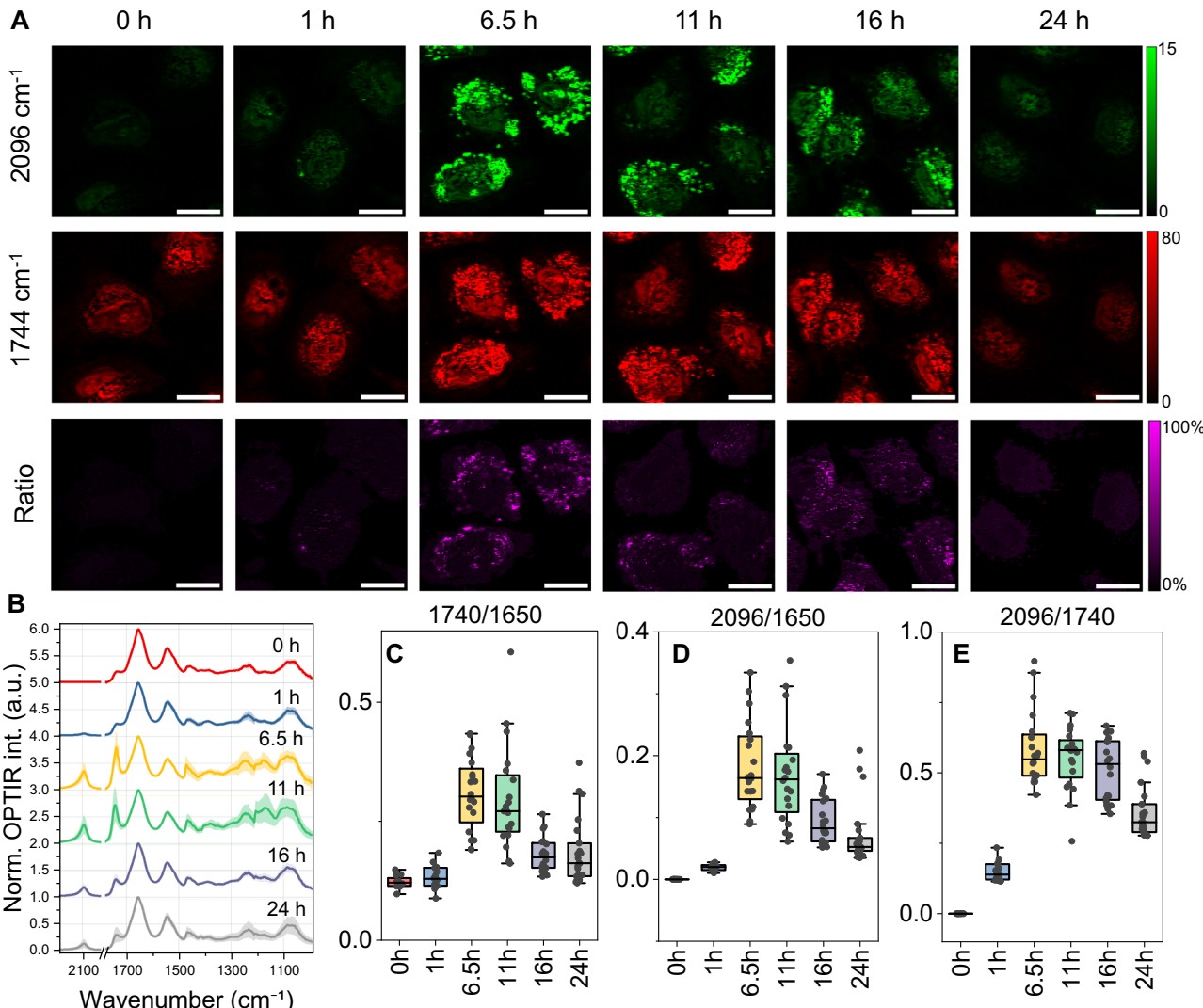

**Fig. 2 | Dynamics of lipid metabolism in cells visualized by OPTIR imaging.**
**A** Neuroglioma H4 cells were incubated with azide-PA and images were collected at 0, 1, 6.5, 11, 16, and 24 h. Representative OPTIR images for newly synthesized lipids (2096 cm$^{-1}$), total lipids (1744 cm$^{-1}$), and the ratioed image of 2096 cm$^{-1}$ to 1744 cm$^{-1}$. Scale bars, 20 μm. **B** OPTIR spectra of cells at corresponding incubation times. Mean and standard deviation are shown in solid curves and shaded areas. Spectra were offset for better visualization. **C–E** Peak ratio calculated from spectral-fitting. Central horizontal lines in the box plot indicate medians, box limits indicate first and third quartiles, vertical whisker lines indicate minimal and maximum values, the outliers were identified using a coefficient of 1.5 times the interquartile range **C–E**. The number of cells used to plot spectra and peak ratio calculation: 0 h ($n = 12$), 1 h ($n = 15$), 6.5 h ($n = 20$), 11 h ($n = 20$), 16 h ($n = 20$), 24 h ($n = 22$). Source data are provided as a Source Data file.

and 2096 cm$^{-1}$ intensities compared to hiPSCs. A strong band more pronounced for the microglia cells around 1170 cm$^{-1}$ can be attributed to the C–O–C stretching mode in lipid ester moieties[61]. This evidence suggests more lipid content can be detected during the progression toward microglial differentiation. Using confocal fluorescence imaging of lipid staining, we observed that microglia without azide-PA supplement already had substantial lipid droplets in cells (Supplementary Fig. 9). For the untreated hiPSCs, the 2096 cm$^{-1}$ signal does not show up in cells, and no peaks were observed when the spectrum was acquired in the cell-free region of the spectrum, indicating the coating of Matrigel to support stem cell growth does not produce an observable signal and therefore would not interfere with the lipid metabolism profiling (Supplementary Fig. 10). The spectral fitting-based quantification (Fig. 3C, D) showed a significantly higher normalized total lipids and newly synthesized lipids to total lipids ratio for hiPSC-derived microglia compared to that of hiPSCs. A similar observation of increased total lipid content was reported for iPSC-differentiated neurons compared to that with iPSCs using spontaneous Raman

spectroscopic measurement[62]. Collectively, these data suggest increased lipid synthesis activities during differentiation, which provides insights into the function of lipids in early brain development[63].

### Distinct lipid metabolism in *GRN*-KD hiPSC & hiPSC-derived microglia

We further tested whether this platform is applicable to studying disease mutation-related metabolic changes. For this purpose, we chose progranulin (PGRN) as our focus. PGRN, encoded by the *GRN* gene, is widely expressed in various tissues including those in the central nervous systems, where it is predominantly found in neurons and microglia[64–66]. PGRN plays a vital role in many physiological processes including regulating lysosomal functions and inflammation. Critically, deficiencies in *GRN* have been linked to a range of neurodegenerative diseases including frontotemporal dementia and Alzheimer's disease[64–66]. *GRN* and PGRN have been closely associated with lipid metabolism. For example, complete loss of PGRN leads to Neuronal Ceroid Lipofuscinosis[67,68], a neurodegenerative disease characterized

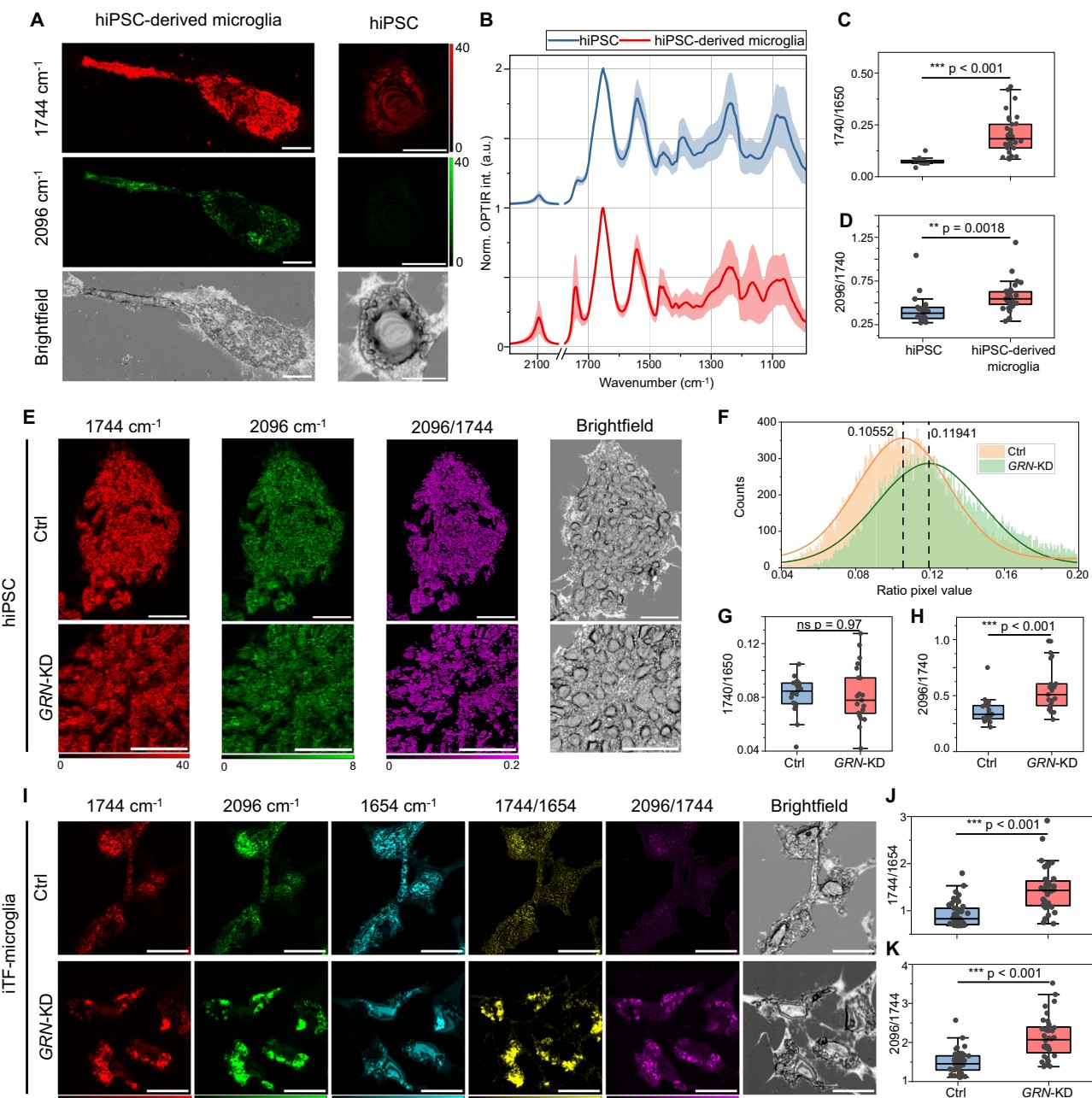

**Fig. 3 | Distinct lipid metabolic levels were observed during hiPSCs microglia differentiation and in *GRN*-KD microglia cells. A** Representative OPTIR images at newly synthesized lipids (2096 cm⁻¹) and total lipids (1744 cm⁻¹) for hiPSC-derived microglia cells and hiPSCs. Scale bars, microglia 20 μm, hiPSCs, 10 μm. **B** OPTIR spectra of hiPSCs and hiPSC-derived microglia cells. Mean (solid curve) and standard deviation (shade) were derived from microglia ($n = 33$) and hiPSC ($n = 23$) cells. Spectra were offset for better visualization. **C, D** Spectral-based quantification of normalized total lipids (1740/1650), and the ratio between newly synthesized lipids to total lipids (2096/1740) from hiPSC ($n = 23$) and microglia ($n = 33$). $p = 9.0e^{-10}$ in **C. E** Representative OPTIR images for control and progranulin knockdown (*GRN*-KD) stem cells acquired at indicated wavenumbers. Ratio images were calculated from 2096/1744 image contrasts. Scale bars, 40 μm. **F** Histogram distribution of ratioed image for control (orange) and *GRN*-KD (green) hiPSCs. Gauss fitting of each group was shown in solid curves with center value shown in black dashed lines. **G, H** Spectral-based quantifications of normalized total lipids and ratio between newly-synthesized lipids to total lipids from control ($n = 21$) and *GRN*-KD ($n = 21$) hiPSCs. $p = 3.2e^{-4}$ in (**H**). (**I**) Representative OPTIR images for control and *GRN*-KD induced-transcription factor (iTF)-microglia cells at indicated wavenumbers and corresponding ratioed images and brightfield images. Scale bars, 20 μm. (**J, K**) Ratioed-image based quantification from control ($n = 46$) and *GRN*-KD ($n = 36$) iTF-microglia cells. $p = 1.3e^{-8}$ in (**J**) and $p = 8.6e^{-10}$ in (**K**). Statistical test: two-sided two-sample t-test (C-D, G-H, J-K). Central horizontal lines in the box plot indicate medians, box limits indicate first and third quartiles, vertical whisker lines indicate minimal and maximum values, the outliers were identified using a coefficient of 1.5 times the interquartile range (C-D, G-H, J-K). Source data are provided as a Source Data file.

by lysosomal accumulation of lipofuscin, a lipid-protein aggregate. Moreover, previous studies have indicated lipid accumulation in humans and mice with *GRN* deficiency, as evidenced through lipid staining and lipidomic methodologies[69,70]. However, there is no direct imaging evidence of lipid metabolism in these systems reported.

Additionally, whether the *GRN* deficiency will impact lipid metabolism as early as in stem cell stages remains unknown. Here, we compared the lipid metabolic levels in control hiPSCs and *GRN* knockdown (KD) hiPSCs. *GRN* expression in *GRN*-KD hiPSCs was significantly reduced based on qPCR analysis (Supplementary Fig. 11). The cells were

incubated with azide-PA at a final concentration of 100 µM for 24 h and then fixed for measurements. OPTIR images of total lipids and newly synthesized lipids were acquired (Fig. 3E). From the histogram distribution of the 2096 cm$^{-1}$/1744 cm$^{-1}$ ratioed images (Fig. 3F), two subgroups exist. By fitting the histogram data with a Gauss function and comparing the center value from the fitting, we observed that the hiPSCs with GRN-KD have higher ratioed values when compared with control hiPSCs, indicating a higher ratio of newly synthesized lipids in GRN-KD cells. To get a quantitative comparison, we performed spectral measurements and spectral fitting (Fig. 3G, H). Consistent with histogram analysis, we observed a significantly higher newly synthesized ratio for GRN-KD cells (Fig. 3H). Interestingly, there is no significant difference in the normalized total lipid (Fig. 3G). We validated the total lipid intensity with fluorescence imaging of cells stained with a lipid marker (Supplementary Fig. 12) and observed a non-significant difference of total lipids in control and GRN-KD hiPSCs, which is consistent with the spectral fitting-based results. The observation of increased newly synthesized lipid ratios and unchanged total lipids in GRN-KD cells may suggest a higher lipid turnover rate in these cells. The significance of directly imaging newly synthesized lipids for precise quantification of lipid metabolism is underscored by this dataset. It cautions against relying solely on fluorescence imaging of total lipids to indicate lipid metabolism, as it may lead to incorrect interpretation.

Building on these insights, we further investigated the implications of GRN-KD in a differentiated cell state. Growing evidence has suggested the pivotal role of microglia in the disease pathogenesis of frontotemporal dementia with GRN mutations, and studies have indicated that Grn$^{-/-}$ microglia accumulate significantly more lipids[70–73]. This led us to investigate if the lipid metabolic alterations observed in hiPSCs were also evident in differentiated microglia cells. Adopting an established protocol[74], we derived microglia from a CRISPRi hiPSC cell line that allows rapid generation of microglia through inducible expression of transcription factors and allows knockdown of endogenous genes. These induced-transcription factor (iTF) microglia-like cells show ramified morphology and express canonical microglia markers (Supplementary Fig. 13). As shown in Supplementary Fig. 14, GRN expression levels were successfully knocked down in GRN-KD microglia cells. The GRN-KD and control iTF-microglia cells were incubated in azide-PA (final concentration 100 µM) containing media for 24 h before being fixed for OPTIR imaging. Representative OPTIR images at 1744 cm$^{-1}$, 2096 cm$^{-1}$, 1654 cm$^{-1}$, and ratioed images are shown in Fig. 3I. An increased contrast was evident in both lipid-associated channels for GRN-KD cells. The ratioed results clearly illustrate GRN-KD cells possess increased levels of normalized total lipids and newly synthesized lipid to total lipids ratio. We further performed statistical analysis comparing the normalized total lipids and the ratio of newly synthesized lipids to total lipids, averaged across single-cell areas, between the control and GRN-KD groups (Fig. 3J, K). Consistent with both imaging results and previous literature[70], the normalized total lipids revealed a significant lipid accumulation in the GRN-KD iTF-microglia when compared to control cells (Fig. 3J). Moreover, the increased newly synthesized to total lipids ratio provides direct evidence of the increased lipid metabolism associated with GRN-KD microglia cells.

Previous studies have indicated high transcriptional similarities between lipid droplet-enriched microglia in Grn$^{-/-}$ mouse brains and lipid droplet-accumulating microglia observed in aged mice[70]. RNA-sequencing (RNA-seq) of Grn$^{-/-}$ mice's lipid droplet-enriched microglia revealed significant upregulation of fatty acid degradation-specific genes, suggesting fortified lipid catabolism in these models. Another lipidomic study revealed that GRN loss leads to an accumulation of polyunsaturated triacylglycerides, as well as a reduction of diacylglycerides and phosphatidylserines in GRN mutant mouse embryonic fibroblasts[69]. This study also performed RNA-seq and identified a panel of lysosomal genes and lipid metabolic genes that are significantly

dysregulated in Grn$^{-/-}$ mouse brains compared to control Grn$^{+/-}$ mouse brains. Further, RNA-seq data from age-dependent microglia in GRN$^{-/-}$ mice indicated significant upregulation of lysosomal functions (Ctsb) and lipid transport (Apoe) genes[72]. These findings indicate an intricate relationship between lipid synthesis, accumulation, breakdown, transport, and GRN deficiency. By conducting further transcriptomic analyses on human-relevant GRN deficiency models employed in the present study, we can directly correlate our phenotypical OPTIR results with genotypic changes. This approach will expand our understanding of the impact of GRN deficiency on lipid metabolism and its potential link to neurodegeneration.

### Cell-type specific lipid metabolic imaging in brain organoids

Different cell types in the central nervous system possess distinct lipid metabolic patterns, which could have implications for various neurological diseases[75]. Therefore, we continued to evaluate the lipid metabolism in the hiPSC-derived brain organoid model system that involves the self-assembly of different cell types including stem cells, progenitor cells, and multiple differentiated cells[76]. To evaluate the lipid metabolism in a human-relevant 3D model system, we cultured brain organoids using the hiPSC line F12442.4. By 5.5 months, the brain organoid had expanded to an average diameter of 3 to 4 mm (Supplementary Fig. 15) and showed an abundant network of organized neurons and astrocytes. 100 µM (final concentration) of azide-PA was added to the culture media and incubated for 24 h. The organoid was then fixed and thin-sectioned to 10 µm for immunostaining. A representative fluorescence image of an organoid section is shown in Fig. 4A where neurons and astrocytes were visualized by staining for TUJ1 (green, neurons) and GFAP (red, astrocytes). To investigate the cell-type specific lipid metabolism, we acquired OPTIR images at 2096 cm$^{-1}$ and 1744 cm$^{-1}$ across organoid sections. A representative FOV is shown in Fig. 4B–D. It is clear that the OPTIR contrast is inhomogeneous across the FOV, and through merged results with fluorescence (OPTIR contrast shown in greyscale), we found that the newly synthesized lipids largely overlapped with the astrocytes. We then performed spectral analysis and observed more pronounced carbonyl and azide signals in astrocytes when compared to neurons (Fig. 4E). Impressively, due to the high spatial resolution of OPTIR, the visualization of small lipid droplets in an astrocyte that was surrounded by neurons, was possible (Fig. 4F). Spectra acquired along a line with fine spacing (1–2 µm) covering neurons and the lipid-containing astrocyte is shown in Fig. 4G. We found that the total lipid and newly synthesized lipid peak intensities increased when spectra were acquired on the astrocytes, but these peak intensities were much reduced when spectra were acquired on neurons. We tested more FOVs and got the Pearson correlation value for OPTIR azide image with fluorescence neuron and astrocyte images (Fig. 4H). A significantly higher correlation between newly synthesized lipids and astrocytes was observed. We further validated our findings with another less diffusive neuronal marker MAP2 and found consistent results (Supplementary Fig. 16). These data indicated that lipid metabolism is heterogeneous across different cell types in hiPSC-derived brain organoids, which coincides with observations in human brain tissue[75]. The higher lipid metabolic levels in astrocytes also provide additional insights into the neuron-astrocytes lipid homeostasis[77].

We further characterized how deep azide-PA can penetrate into the brain organoid. We took fluorescence images and OPTIR images at different depths from the surface of an organoid slice (Fig. 4I–K). It is clear that the azide signal is concentrated on the surface of the organoid and gradually decreases when moving toward the core. We took spectra at different locations and observed a decrease in total lipids and newly synthesized lipids when moving inwards (Fig. 4L). The quantification of newly synthesized lipids was performed by integrating the area under the curve to fit a 2096 cm$^{-1}$ peak and plotting it as a function of distance from the surface (Fig. 4M). Consistent with the

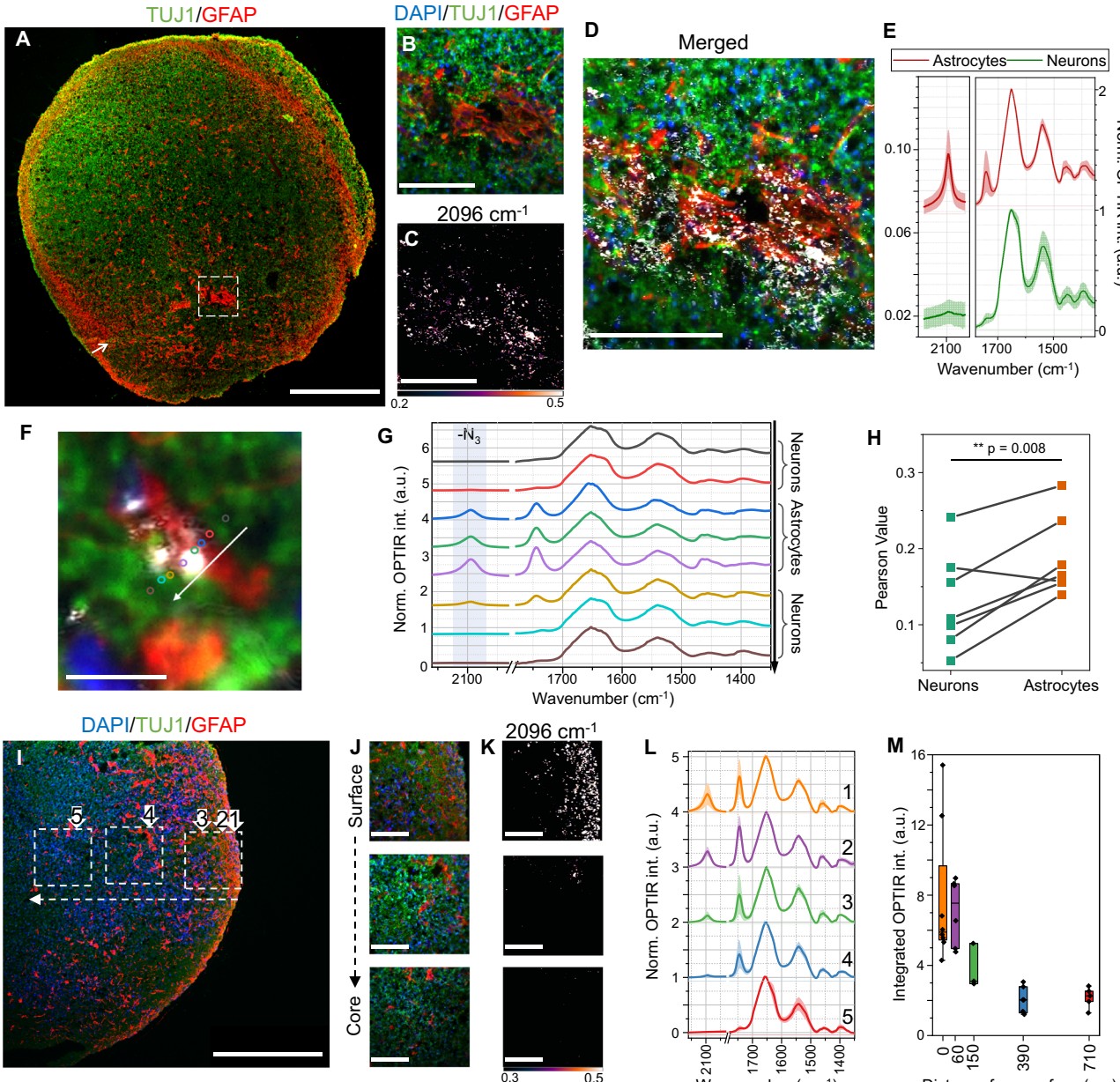

**Fig. 4 | Cell-type specific lipid metabolic imaging in hiPSC-derived brain organoids. A** Immunofluorescence imaging showed the distribution of neurons (TUJ1) and astrocytes (GFAP). Scale bar, 500 µm. **B–D** Zoom-in view of the dashed area indicated in **A**. Scale bars, 100 µm. In the merged image **C**, OPTIR contrast is shown in greyscale. **E** Mean (solid curve) and standard deviation (shade) of spectra acquired on neurons (n = 7) and astrocytes (n = 10). **F, G** Spectral acquisition across a small FOV (white-arrow indicated in **A**). Location for spectral measurements is indicated with circles that have the same color coding for spectra. Scale bar, 10 µm. **H** Pearson correlation value showed higher colocalization of newly synthesized lipids with astrocytes than neurons. Each pair represents an FOV with dimensions of 200 to 390 µm square (n = 7). **I–K** Fluorescence and OPTIR azide imaging at varying distances from the surface of an organoid slice. The location of the three fields of view in **J, K** (top to bottom) is indicated in (**I**) (right to left) with white dashed squares. The surface-to-core direction is also indicated with dashed arrows. Scale bars, 500 µm in **I**, 100 µm in **J, K**. **L** OPTIR spectra at different distances from the organoid surface, with the location indicated with numbers 1–5 in **I**. Mean (solid curve) and standard derivation (shade) were generated from 8 (Location 1), 6 (Location 2), 3 (Location 3), 7 (Location 4), and 5 (Location 5) cells. **M** Integration of the intensity of 2096 cm$^{-1}$ peak as a function of distance to surface. The data generated is based on the spectral measured in **L**. For spectra shown in **E**, **G**, and **L**, all raw spectra were normalized to the intensity at 1654 cm$^{-1}$ and offset for better visualization. Statistical tests: two-sided paired sample t-test **H**. Central horizontal lines in the box plot indicate medians, box limits indicate first and third quartiles, vertical whisker lines indicate minimal and maximum values, the outliers were identified using a coefficient of 1.5 times the interquartile range **M**. Source data are provided as a Source Data file.

imaging results, we observed a gradual decrease of 2096 cm$^{-1}$ peak intensity with detectable signal around 700 µm from the surface.

Given that brain organoids lack a vascularization system, their interior cells are subjected to hypoxia and necrosis from restricted oxygen and nutrient delivery by surface diffusion[76,78]. This leads to a denser cell population at the periphery. Consequently, the outer cells are naturally the primary absorbers of azide-PA in the cell culture medium, potentially limiting the compound's diffusion into deeper layers. This establishes an azide-PA concentration gradient, with higher levels near the surface and decreasing concentrations toward the core. Previous research has suggested that the viable region of organoids is typically limited to a few hundred µm from the surface, despite efforts to enhance oxygen and nutrient diffusion[79–81]. Given this context, it is possible that metabolism towards the core is compromised due to

hypoxia and nutrient scarcity. Therefore, the observed reduction of OPTIR azide signal towards the core is likely a complex interplay between cell density, azide-PA concentration gradient, and potential metabolic alterations due to oxygen and nutrient deficiency. In future studies, we aim to integrate organoid slices into our metabolic research. These slices can access oxygen and nutrients from both top and bottom surfaces and have shown to substantially reduce internal cell death[81].

## Discussion

In this study, we reported a single-cell metabolic imaging platform that enables direct imaging of lipid metabolic activity in human-relevant model systems with high resolution and high specificity. The azide labelled PA was used as a metabolic tracer to study the lipid metabolism in various systems under different conditions. Although we used PA as a testing bed in this report, the platform can be easily adapted to study the metabolism of other molecules. For example, cholesterol has been implicated in the pathogenesis of many neurodegenerative diseases[82], and the developed platform could be utilized with commercially available azide-tagged cholesterol analogs to investigate cholesterol distribution in cells as well as its interaction with other biomolecules, thereby providing new insights into the role of cholesterol in cellular processes. Moreover, to expand the IR probe library, it will be valuable to test and compare other IR tags, such as nitriles, to the azide tag used in this study. By doing so, we can further enhance the capability of the platform to investigate a broader range of metabolic processes and pathways.

It is worthwhile to compare the reported metabolic imaging platform with other single-cell optical metabolic imaging methods. Optical metabolic imaging (OMI) employs the fluorescence imaging of cellular endogenous co-enzymes that are involved in metabolism, providing a map of general metabolic activities in a label-free manner[83]. Due to the signal contrast mechanisms used in OMI, it is challenging to apply this technique to study specific metabolic pathways of a target molecule such as cholesterol. Additionally, photobleaching may occur in OMI measurements, which can lead to inconsistent fluorescence intensity or lifetime quantifications. Another metabolic characterization technology based on vibrational spectroscopy is Raman scattering. While single-cell spontaneous Raman spectroscopy provides full spectral coverage, the presence of strong fluorescence background overlapping with Raman scattering wavelengths may complicate spectral analysis[84,85]. Furthermore, it does not support single-color imaging at the wavelength of interest, which limits its throughput. Stimulated Raman scattering (SRS) microscopy provides high speed imaging at specific wavelengths with narrow wavelength coverage[86]. The instrumentation of the SRS system is complex as it requires spatial and temporal overlap of two invisible near-IR lights, and they are more expensive, as ultrafast lasers are required. In contrast, the OPTIR signal does not suffer from fluorescence background, and it can be easily switched between single-color imaging mode and single-point full wavelength coverage mode. Furthermore, the OPTIR setups stand out by not requiring ultrafast lasers, which significantly reduces both the complexity and cost.

To fully explore the potential of OPTIR metabolic imaging platforms for biomedical discovery, we aim to improve the performance of the current system. To provide statistical robustness, the capability to acquire data from a large number of cells in a reasonable time frame is essential. In the present work, our analysis based on tens of cells was sufficient to suggest a statistical difference between experimental groups. However, more cells may be needed to reveal more subtle biological differences or reveal heterogeneity. For example, cellular experiments that highlight the heterogeneity typically use $10^3$ to $10^5$ cells[87,88]. In this study, acquiring a single wavenumber OPTIR image with a size of 280 by 280 μm and a step size of 0.5 μm takes about 330 s. Assuming the average size of a single cell is 40 μm in diameter,

this corresponds to about 60 cells in the 280 by 280 μm FOV. This translates to about 4.5 h data acquisition for $10^3$ cells covering 3 IR wavenumbers. This low throughput will also make full-hyperspectral imaging impractical for a large number of cells. To improve imaging speed, multiple methods from hardware to software innovations are envisioned. Widefield OPTIR setups can be utilized[89,90], where the imaging speed ranges from a few to tens of Hz, leading to a significantly reduced hyperspectral imaging time to tens of minutes. Sparse sampling in the spectral domain can be used in tandem with widefield OPTIR setup to further boost the hyperspectral imaging speed[91]. For point-scan based OPTIR setups, digitization of OPTIR signals using a fast digitizer could improve signal-to-noise (SNR) and thus improve the imaging speed[92]. Video-rate OPTIR imaging has recently been achieved by coupling the laser-scanning geometry[93] or by optimizing the key parameters such as mid-IR laser and widefield detectors[94]. Additionally, advancement in denoising methods to recover the images acquired at high-speed with low SNR could also be incorporated to improve the overall OPTIR hyperspectral imaging speed[95]. Besides imaging throughput, another endeavor to pursue is the spatial resolution of the current OPTIR setups. Although achieving much-improved spatial resolution for IR imaging, the resolution is still diffraction-limited for the visible beam. Recent advances in breaking the diffraction limit of the visible beam through detection of higher harmonics[96] or illumination at multiple angles[97] to improve the frequency-domain coverage, both achieve around 120 nm resolution. We can easily adapt these modalities into our OPTIR metabolic imaging system and achieving improved resolution.

While the presented work utilizes fixed cells and tissue as a testing bed to demonstrate the feasibility, the platform can be readily translated to study live-cell lipid metabolism. Reports utilizing isotopes to study lipid metabolism and glucose metabolism in live cells have been achieved with OPTIR setups[41,42,98]. By comparing the spectra and ratios between $^{13}C$ and $^{12}C$ ester carbonyls of live cells and fixed cells, Shuster et al. concluded the fixed cells can provide a reliable representation of de novo lipogenesis as observed in live cells[98]. Spadea et al. have reported the OPTIR spectral comparison between live and fixed cancer cell lines and observed similar protein amide I to amide II ratios in live and fixed cells[99]. These findings suggest that fixed cells can be a good representative of live cells when studying lipid metabolism. However, the fixation process can modify the cell morphology and cause cell shrinkage[100], thus introducing unwanted spatial overlap of targeted molecules and other interfering molecules within the diffraction-limited spots[98]. Therefore, to better preserve cell morphology and enable longitudinal studies of lipid metabolism, live-cell imaging will be a preferred option. Additionally, by implementing the throughput-optimizing methods described above, potential motion artifacts that could complicate quantifications in live-cell imaging can be minimized. Another note on live-cell imaging is the localized temperature increase may raise concerns about interfering with normal cellular functions. We can estimate the thermal effect based on the modulation depth of the OPTIR process. Take a cell image as an example, the DC signal is around 3.5 V, and the AC signal is 20 mV, with the AC gain of 2 times, we can calculate the modulation depth to be $2.9 \times 10^{-3}$. Since the AC signal is generated based on scattering intensities difference between IR-on and IR-off states, and studies have established the dependence of scattering intensity on temperature to be $10^{-3}/K$[92,101], we can estimate our localized temperature increase is around 2.9 K. Additionally, this temperature increase is transient and lasts shorter than the IR pulse width of 1 μs. The calculated temperature increase scale is consistent with other photothermal IR reports where different readouts such as fluorescence intensity fluctuation (2–4 K)[102] and quantitate phase (0.1 K)[103]. Therefore, with the current configuration, we do not expect the thermal effect to significantly impact cellular normal physiology. However, if it is a concern, the local temperature can be reduced by reducing the IR power or shortening the IR pulse width.

With the metabolic imaging feasibility demonstrated, the developed platform is ready for application to a wide range of metabolic-related questions. In this study, we identified an elevated lipid metabolism in *GRN*-deficient hiPSCs and hiPSC-derived microglia cells. Moving forward, we aim to explore if similar phenotypes are evident in other cell types such as *GRN*-KD astrocytes, which have been shown to promote synaptic dysfunction[71,104]. To deepen our insights, we plan to incorporate more complex systems such as hiPSC-derived brain organoids and animal models[69,70,104]. Through these investigations, we aim to explore how *GRN* deficiency differentially impacts lipid metabolism across cell types, brain regions, and model systems. These insights could potentially shed light on the underlying mechanisms of neurodegenerative diseases linked to *GRN* mutation and offer new avenues for therapeutic interventions. Extending from the *GRN* gene, we can study more broadly lipid accumulation related to aging and neurodegeneration[105], which have been widely observed but lack the underlying mechanism to explain the phenotype. Benefiting from the superior resolution, our platform can also be applied to study metabolic heterogeneities in microorganisms, where individual cell metabolic responses can be mapped and potentially employed as a screening tool to optimize biofuel energy production[106]. Collectively, we anticipate OPTIR microscopy, together with sensitive IR probes, will transform the single-cell metabolic imaging field, enabling more profound biological discoveries and improved disease treatment.

## Methods

### Ethical statement
Our research complies with all relevant ethical regulations at the University of California Santa Barbara and study procedures were approved by the Institutional Review Board (#767).

### Human neuroglioma cell culture and treatment
H4 (male neuroglioma, ATCC HTB−148) cells were cultured in DMEM supplemented with 10% FBS and 100 μg/ml penicillin/streptomycin. Cultures were maintained at 37˚C, 5% $CO_2$, and passaged every 3-4 days when 80% confluency was reached. 10-mm diameter $CaF_2$ slides were placed in each well of the 24-well plate and were soaked in 70% ethanol for 15 min. Before cells were plated, the slides were rinsed with MilliQ water twice. H4 cells were plated at 67,500 cells per well in the 24-well plate the day before treatment. On the day of treatment, cells were 50% confluent. Azide palmitic acid (Click Chemistry Tools, 1346) or palmitic acid (Sigma Aldrich, P0500) was added directly to the media at the final concentration of 100 μM. The slides with treated cells were collected after 0, 1, 6.5, 11, 16, and 24 h. The slides were rinsed with cold PBS once, incubated in 4% PFA at room temperature for 10 min, and then rinsed again with cold PBS twice. Before OPTIR imaging, cells were rinsed with MilliQ water twice to remove the excessive salt deposit. The cells were then air-dried and ready for OPTIR imaging and spectral measurements. For Triacsin C treatment, the stock chemicals were diluted in cell culture media to a final concentration of 1 μM. 100 μM azide-PA was added to the cell culture media without or without Triacsin C, and cultured 24 h before collection.

### hiPSC culture for *GRN* knockdown and microglia differentiation
The hiPSC lines used in this study were CRISPRi hiPSC[107], an iPSC line with a constitutive CRISPRi machinery (dCas9-BFP-KRAB), and iTF-hiPSC[74], an iPSC line expressing inducible CRISPRi machinery and inducibly expressing six transcription factors that enable the generation of microglia-like cells. Both hiPSC lines were generated in the background of the WTC11 human iPSC line (male, Coriell Catalog No. GM25256), and were kindly provided by Dr. Martin Kampmann (UCSF). HiPSCs were cultured in mTeSR Plus medium (StemCell Technologies) in 6-well plates (Corning) coated with Matrigel (Corning). Medium was changed daily, and cells were passed with ReleSR (StemCell Technologies) when 70-80% confluent.

### Knockdown of *GRN*
To knockdown *GRN*, a sgRNA targeting *GRN* was lentivirally packaged in HEK293T cells, and transduced into CRISPRi- or iTF-hiPSCs. The control hiPSCs were transduced with a scrambled sgRNA. Cells were selected with 2 μg/ml puromycin (Gibco). To quantify *GRN* knockdown, total RNA was extracted from iPSC using PureLink RNA Mini Kit (ThermoFisher Scientific), and cDNA synthesized with the ProtoScript® II First Strand cDNA Synthesis Kit (NEB). qPCR was performed on an Applied Biosystems QuantStudio 6 Pro Real-Time PCR System, using TaqMan probes (ThermoFisher Scientific) specific for *GRN* (Hs00963707_g1) and for *PPIA* (endogenous reference, Hs99999904_m1). Expression fold change was calculated using the ΔΔCt method.

### hiPSC-derived microglia differentiation
Microglia cells were differentiated from hiPSC following a published protocol[60]. Briefly, hiPSC were allowed to grow until 90-95% confluency. From day 0 (D0) to D3, cells were fed daily with mTeSR plus supplemented with 80 ng/ml BMP4. On D4, medium was replaced with StemPro-34 SFM medium as a base containing the following: 2 mM GlutaMax (Gibco), 25 ng/mL bFGF, 100 ng/mL SCF and 80 ng/mL VEGF. On D6, medium was replaced with StemPro-34 SFM supplemented with 50 ng/mL M-CSF, 50 ng/mL SCF, 50 ng/mL IL-3, 5 ng/mL Thrombopoietin and 50 ng/mL Flt3 ligand. On D10, supernatant was collected, cells were pelleted and resuspended in the same medium as D6. Every 4 days, from D14 until D25-D50, cells from the supernatant were pelleted then resuspended in StemPro-34 SFM supplemented with 50 ng/mL M-CSF, 50 ng/mL Flt3l and 25 ng/mL GM-CSF, and returned to the same well. Around D25, microglial progenitors floating in the medium were collected, pelleted and resuspended in microglia differentiation medium containing RPMI−1640 (Gibco), plus 2 mM GlutaMAX, 10 ng/ml GM-CSF and 100 ng/ml IL-34. Cells were plated into a new 6-well plate, and microglia was matured in the same media for 14 days, with full media change every 4 days. Cells were assayed at D14. All cytokines were purchased from Peprotech.

### iTF-microglia differentiation
Induced-transcription factor (iTF) microglia-like cells were generated following a published protocol[74]. Briefly, iTF-iPSC were allowed to grow until 90−95% confluency. On D0, iPSC colonies were dissociated as single cells with Accutase (StemCell Technologies) and replated on Matrigel-coated 6-well plates in mTeSR Plus medium (StemCell Technologies), 10 nM ROCK inhibitor, and 2 ug/mL doxycycline (Sigma). On D2, medium was replaced with Advanced DMEM/F12 Medium as a base, plus 1× GlutaMax (Life Technologies, 35050), 1× Antibiotic-Antimycotic (Gibco), 2 ug/mL doxycycline, 10 ng/ml GM-CSF (Peprotech), 100 ng/ml IL-34 (Peprotech), and 50 nM TMP (MP Biomedical) to induce CRISPRi activity. On D4, cells were fed with the same medium as D4, and supplemented with 50 ng/ml M-CSF (Peprotech) and 50 ng/ml TGFB1 (Peprotech). Medium changed every 2 days, and iTF-microglia were assayed at D10.

### Immunocytochemistry of hiPSC-derived microglia and hiPSC
For immunofluorescence, hiPSC and hiPSC-derived microglia/iTF-microglia were fixed with 4% PFA for 10 min at room temperature (RT), then washed 3× with PBS. Fixed cells were incubated in blocking buffer containing PBTA (0.5% BSA and 0.1% Triton X−100 in PBS) plus 5% normal donkey serum for 1 h at room temperature. Samples were then incubated with primary antibodies in blocking buffer overnight at 4 °C, washed three times with PBTA, then incubated with secondary antibody in blocking buffer for 1 h at room temperature. Samples were washed three times with PBTA, then mounted with Prolong Diamond

Antifade Mountant (Invitrogen) with DAPI for nuclei staining. Antibodies used were 1:1000 rabbit anti-IBA1 (Wako, 019-19741), 1:500 rabbit anti-CD45 (Abcam, ab214437), and 1:1000 donkey anti-rabbit Alexa Fluor 488 (Invitrogen, A-21206).

### Brain organoids culture
The hiPSC line F12442.4[108] was kindly provided by Celeste Karch and used for organoid experiments. iPSCs were cultured in mTeSR Plus medium (Stem Cell Techonologies) on tissue culture plates coated with hESC-qualified Matrigel (Corning). mTeSR Plus was exchanged every other day and iPSCs were routinely passed using ReLeSR (Stem Cell Technologies). Cerebral organoids were generated using microfiber scaffolds as described in Lancaster et al.[109], with the modification that no Matrigel was added after day 40 of differentiation to medium IDM + A (1:1 of DMEM/F12 and Neurobasal, 0.5% N2 supplement, 2% B27 +vitamin A, 0.25% insulin solution, 50 μM 2-mercaptoethanol, 1% Glutamax, 0.5% MEM-NEAA, 1% penicillin-streptomycin, 0.4 mM vitamin C, and 1.49 g HEPES per 500 ml).

### Brain organoid tissue preparation
5.5-months-old organoids were incubated in 100 μM azide palmitic acid in IDM + A medium for 24 h. Organoids were washed 2x in IDM + A media, 2× in PBS, and fixed in 4% PFA at 4 °C o/n. Following fixation, organoids were washed 3x in PBS for 15 min and cryoprotected in 30% sucrose at 4 °C o/n. Organoids were embedded in NEG-50 media, frozen and cryosectioned on a Cryostat (Leica CM1850) at 10 μm thickness. Sections were collected on Superfrost Plus Microscope Slides (Thermo Fisher).

### Brain organoid tissue immunohistochemistry
Organoid sections were washed in PBS for 5 min, permeabilized in 0.3% Triton-X in PBS for 5 min and incubated in blocking solution (10% normal goat serum, 1% bovine serum albumin, 0.3% Triton-X in PBS) for 30 min at RT. Then, sections were incubated in primary antibodies (mouse anti-Tuj1 at 1:500, Sigma T8578; chicken anti-GFAP at 1:1000, abcam ab4674) in blocking solution at 4 °C o/n. After washing 3× for 5 min in 0.1% Tween in PBS (PBT), sections were incubated in secondary antibodies (Alexa Fluor 488-labeled goat anti-mouse antibody, Thermo Fisher A11001; Alexa Fluor 647-labeled goat anti-chicken antibody, Life tech A-21449), both at 1:500 in PBT, for 1 h at RT. For MAP2 and GFAP staining, organoid sections were washed in phosphate buffer (PB) 3 x for 5 min on a rocker, permeabilized in 1% Triton-X in PB (PBT) 2x for 5 min on the rocker and incubated in blocking solution (1% bovine serum albumin, 1% Triton-X in PB) for 1 h at RT. Sections were incubated in primary antibodies (rabbit anti-MAP2 at 1:750, Thermo Fisher PA5-17646; chicken anti-GFAP at 1:1000, Abcam ab4674) in antibody solution (0.3% bovine serum albumin, 1% Triton-X in PB) at RT for 2 h, on a rocker. After washing 3x for 5 min in PBT, sections were incubated in secondary antibodies (Alexa Fluor 488-labeled donkey anti-rabbit antibody, Thermo Fisher A21206; Alexa Fluor 647-labeled goat anti-chicken antibody, Life tech A-21449), both at 1:500 in PBT, for 1 h at RT. Finally, sections were washed (2 × 5 min in PBT, 1 × 5 min in PBS), stained with 0.5 μg/ml DAPI for 10 min at RT and washed 3× for 5 min in PBS. For OPTIR imaging, sections were rinsed with MiliQ water twice and then air-dried.

### Fluorescence-integrated OPTIR setup
OPTIR imaging was performed on mIRage LS (Photothermal Spectroscopy Corp.). The system consists of a mid-IR pump beam and a visible probe beam. The mid-IR beam was a pulsed quantum cascade laser running at 100 kHz repetition rate and 1 to 10% duty cycle. The visible probe was a continuous wave laser with a center wavelength of 532 nm. Two geometries were used for the measurement: counter-propagation and co-propagation of the IR and visible beam. For counter-propagation, the mid-IR beam was focused below the sample substrate (CaF$_2$)

with a reflective objective (40x, 0.78NA, Pike Technologies), and the visible beam was focused from above the sample substrate with a refractive objective (50x, 0.8NA, Olympus). For co-propagation, both the mid-IR and visible beam was focused with the reflective objective. Epi-detected light was collected and focused on a photodiode. The OPTIR signal was demodulated with a lock-in amplifier and the image was created by raster scanning with an XY motorized stage. The system was enclosed and purged under gentle nitrogen flow to minimize water vapor interference of spectra. For widefield fluorescence imaging, filter cube sets covering common fluorophore and fluorescence proteins in blue, green, red, and far-red channels were used. A camera with a high quantum yield was used to capture the fluorescence images.

### OPTIR image, spectral acquisition, and fluorescence imaging
The OPTIR images of Fig. 1D and Fig. 3I were acquired with a step size of 0.2 μm, while the image in Fig. 3A was acquired with a step size of 0.15 μm. Other data presented were acquired with 0.5 μm pixel size. The acquisition time for an OPTIR image varied from tens of seconds to around 5 min depending on the FOV, pixel size, and stage scan rates. The recipe used for OPTIR spectral measurement was 200 cm$^{-1}$/s scan speed. The spectral scan range was 980 to 2300 cm$^{-1}$ for neuroglioma cells, stem cells, and microglia cells. 1300 to 2300 cm$^{-1}$ was scanned for brain organoid samples to avoid strong glass absorption below 1300 cm$^{-1}$. 1780 to 2030 cm$^{-1}$ spectral range was omitted for all spectra shown in the manuscript due to no observable peaks presented in the range. Fluorescence imaging of the whole brain organoid tissue section was performed by stitching 4 images acquired with a 10x objective. Exposure time for fluorescence imaging was 100 μs for the DAPI channel, 1 s for TUJ1 channel, and 1 s for GFAP channel. The IR power at the sample was 5 mW for co-propagation measurement of brain organoid tissue samples, and 6 mW for counter-propagation measurement for rest of samples. IR power was measured at 2096 cm$^{-1}$. All OPTIR spectra shown in the manuscript were normalized with the IR laser power spectrum. The visible power at the sample was in the range of 2.5 to 16.3 mW depending on the sample.

### OPTIR spectra and image analysis
Spectra analysis, image analysis, and quantifications were performed with OriginPro 2022b (OriginLab Corporation) and ImageJ if otherwise noted. For the normalized (Norm.) spectra shown in the manuscript, raw spectra were normalized to total protein content at 1654 cm$^{-1}$. For Fig. 1C, PA and azide-PA powder OPTIR spectra were normalized to 1700 cm$^{-1}$ and the cell spectrum was normalized to 1654 cm$^{-1}$. To derive a spectrum from a single cell, we collected and averaged at least 3 pinpoint spectra after normalization. This resultant spectrum is used for spectral fitting and area under curve calculations. For spectral-based quantifications, whole spectral fitting was used with the peaks found using smoothed second derivatives (Peak Analyzer function in OriginPro, detailed parameters listed in the caption of Supplementary Fig. 5). The calculation of the ratio between newly synthesized lipids, total lipids, and protein was done using the area under the curve from the fitted results. OPTIR images were normalized to the IR power intensity at the corresponding wavenumber. For the Ratioed image presented in Figs. 2A, 3E, I, the build-in function 'ratio' of the OPTIR data acquisition software PTIR Studio 4.5 (Photothermal Spectroscopy Corp.) was used. For image correlation analysis presented in Fig. 4H, the corr function in MATLAB 2021b (MathWorks) was used to generate the Pearson correlation coefficient.

### Statistics & reproducibility
All statistical analysis were performed using OriginPro 2022b software (OriginLab Corporation). All experiments were repeated at least 3 times. Data were analyzed for statistical significance using two-sided

**Article**

two-sample t-test or two-sided paired sample t-test. $p$ values < 0.05 were considered statistically significant.

## Reporting summary

Further information on research design is available in the Nature Portfolio Reporting Summary linked to this article.

## Data availability

All data supporting the findings of this study are presented in this manuscript and the Supplementary Information. All the raw data has been provided in the Source Data Files with no restrictions to access. Further information and requests for resources can be directed to the corresponding authors, and requests will be fulfilled within 6 weeks.

## Code availability

The code-based analysis is done following standard guidelines of the software functions. User-defined parameters are provided in the Methods section. No custom code is developed or used.

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

## Acknowledgements

This research was funded by grants from the NIH NINDS U54 NS100717-01 (K.S.K.), the W.M. Keck Foundation (K.S.K.), and the Alzheimer Association (K.S.K.). We thank Celeste Karch (Washington University) for kindly providing the hiPSCs used to produce brain organoids in this study with funding support NIH P30 AG066444 and Rainwater Charitable Organization. The recruitment and clinical characterization of research participants at Washington University were supported by NIH P50 AG05681, P01 AG03991, and P01 AG026276. We also thank Dr. Martin Kampmann for providing CRISPRi and iTF-iPSCs. We acknowledge the use of cryostat in NRI-MCDB Microscopy Facility at the University of California, Santa Barbara. We appreciate Dr. Caitlin Davis from Yale University for kindly allowing us to use the OPTIR setup for additional experiments. Y.B. is grateful for the fruitful discussion with Dr. Ying Jiang from the Massachusetts Institute of Technology. Partial figure drafts were prepared using Biorender.com.

## Author contributions

Y.B. conceived the idea, designed the project, performed the imaging experiments, and analyze the data. C.M.C. performed hiPSC culture, microglia differentiation and characterizations, lipid staining, and fluorescence imaging of hiPSCs. S.M.K.G. performed the brain organoid culture and tissue preparation. S.M.K.G. and R.G. performed the brain organoid tissue immunohistochemistry. X.T., Y.B., R.G. and A.P.L. performed H4 cell culture, drug treatment, and lipid droplets staining. All authors discussed the results. Y.B. wrote the manuscript with the inputs from other co-authors. K.S.K. supervised the project.

## Competing interests

K.S.K. consults with Expansion Therapeutics, ADRx, Herophilus and serves on the BOD of the Tau Consortium and Minerva Biotechnologies. Y.B. is a part-time contractor for Photothermal Spectroscopy Corp. The remaining authors declare no competing interests.
