## [Peer Review File · Nature Communications]

REVIEWERS' COMMENTS:

Reviewer #1 (Remarks to the Author)

In their manuscript "Sub-cellular mapping of lipid metabolites using IR probe in human-derived model systems" Yeran Bai et al., report the application of Optical Photothermal Infrared (OPTIR) microscopy in combination with a mid-IR tag, Azide. The application of Azide as vibrational probe facilitates chemical imaging of lipid-fate by improving absorption cross-section over other vibrational probes, such as glucose-d7, by achieving spectral clearance in the spectral silent region of biomolecules. Although the imaging method has been broadly exposed previously and the concept of mid-IR Tags in the silent region has been explored and reported before, I found the combination of OPTIR with Azide a quite promising and timely approach in order to further enhance sensitivity in optothermal imaging (or mid-IR imaging in general) and to expand the library of mid-IR probes. I would like to congratulate the authors on this innovative approach. Overall, the manuscript is well-written, easy to follow and the results are presented with clarity in support of the conclusions. I would like to endorse its publication in Nature Communications as it could be of interest to its broad audience particularly for biotechnology and microscopy communities.

Nevertheless, there are a couple of remarks that might help the authors to improve the work further.

1-I would be very curious to see the limit of detection of OPTIR for Azide, as well as the limit of the detection for the tagged lipids, PA for instance. They report adding 100 μ M azide-PA in their neuron/astrocytes experiments, what is the final concentration after adding to the media? What was the azide-TAG concentration for the other experiments? and, more importantly, how do these concentrations compare with the limit of detection? how many times above or below?

2-It is not clear how many cells were arbitrarily selected for their single-cell spectral analysis; this information might be key to support statistical relevance (with biologists used to measure 10's of thousand to millions of cells).

3-Since focused pump-probe beams were used, this might signify elevated optical/thermal effects to the cells. Have the authors studied the effects of tight irradiation on the cells?

4-The text is not explicit on whether the cells were alive or fixed; although, by looking at the methods, and considering the imaging speed, it becomes clear that the cells/tissues were fixed. I would appreciate if the authors could clearly state in the script whether the cells are fixed or alive, and (to avoid ambiguity) discuss how it could be applied in live-cell microscopy if not. This is relevant as the use of Azide and OPTIR might become of interest for biologists in longitudinal studies.

Reviewer #2 (Remarks to the Author)

This manuscript titled "Sub-cellular mapping of lipid metabolites using IR probe in human-derived model systems" by Yeran Bai and colleagues demonstrates the use of recently developed imaging technology optical photothermal infrared (OPTIR) microscope in detecting and analyzing metabolic heterogeneity, specifically in lipid metabolism, among different cell types. First, the authors revealed that the OPTIR microscope coupled with supplementation of azide-tagged fatty acid (azide-PA) is capable of detecting newly synthesized lipids in an immortalized cell line, specifically human neuroglioma H4 cells. Next, they examined lipid metabolism in hiPSCs and hiPSC-derived microglia and showed that compared to hiPSCs differentiated microglia have significantly higher total lipid and newly synthesized lipid levels. In parallel, they also found that progranulin (PGRN) knockdown in hiPSCs leads to an increase in newly synthesized lipids without altering the level of total lipids, thus to a faster lipid turnover rate. Finally, they utilized the OPTIR microscope to investigate the lipid metabolism in the hiPSC-derived brain organoid model system and showed that lipid droplets containing newly synthesized lipids are found in astrocytes, and not in neurons.

Overall, they demonstrate the use of cutting-edge imaging technology, OPTIR microscope, with several different human-relevant model systems. There is a vast potential in the use of OPTIR metabolic imaging and this manuscript demonstrates some of them, specifically focusing on lipid metabolism. Although they do not reveal a novel biological finding, they open up new avenues for further investigation of lipid metabolism in different cell types, as well as the metabolism of distinct metabolites using different biorthogonal IR tags to assess the metabolic heterogeneity among cells and tissues.

Some of their experimental results need major and minor revisions which are listed below.

Major comments:

1. The authors choose to use the azide tag, which has a peak in the cell-silent region, which is great. Does this tag affect the fatty acids to be incorporated into phospholipids? We do not see many signals coming from the plasma membrane. Using OPTIR imaging, is it possible to image membrane incorporation of lipids?
2. Figure 1D shows the signal coming from newly synthesized lipids from a single cell. A larger image would have been better. It is not possible to see individual lipid droplets. Also, what type of cell did they use for this experiment? Did the authors co-stain the cells with BODIPY or any other lipid droplet marker to show that the signal is coming from lipid droplets?
3. In Figure 1E, there is a clear intensity difference in OPTIR signal at 1744 cm^{-1} between azide-PA incubated and PA incubated cells. If this signal is coming from total lipids, how did the authors explain the increase in total lipids with azide-PA incubation? A change in total lipids with azide-PA feeding could be misleading for future analyses.
4. This question follows up on the previous one. The concentration of azide-PA is 100 μM . Too much PA is known to be toxic to cells. Did the authors try different concentrations? Maybe lower azide-PA concentrations which do not cause an increase in total lipids should be examined.
5. For Triacsin C treatment, the authors only included spectral-based quantification. It would be better to include some representative OPTIR images.
6. For the time-dependent incorporation of the azide experiment (Figure 2), did the authors use the neuroglioma cells or a different type of cell? Again, why did they observe an increase in total lipids, along with azide-PA levels which indicate newly synthesized lipids? Also, why did the total lipid and newly synthesized lipid levels decrease after 16 hours, if there is a continuous supply of azide-PA in the culture? By 24 hours, both signals were very low. Total lipid levels should not be affected by azide-PA incorporation and if azide-PA is in the culture for the whole 24 hours, the cells should continue to synthesize new lipids.
7. Figure 3A shows the representative OPTIR images of newly synthesized lipids in hiPSC and hiPSC-derived microglia. It would be better to have bigger images. It is a little surprising to see such a high level of total lipids and that many lipid droplets in microglia. Can the authors comment on this? They mention about them in the main text as "lipid droplet-like structures". Did they co-stain the cells with any lipid droplet marker or other organelle markers to visualize where OPTIR signals are coming from? Interestingly, not all the "lipid droplet-like structures" contain newly synthesized lipids, as seen in the 2096 cm^{-1} image. And the ones that have newly synthesized lipids are localized more in the periphery, rather than perinuclear. Is this the case for this specific image or for all?
8. The finding that total lipids in PGRN-KD cells do not change, but they have a higher lipid turnover rate indicated by increased levels of newly synthesized lipids, is very exciting. If those cells have faster synthesis, they should also have faster lipid catabolism. Is this known from transcriptomic studies? Can the authors elaborate on this finding more in the main text?
9. Following up on the previous comment, in Figure S4, the resolution of the confocal image of

cells is very low. Also, it seems the LipidTox staining was very heterogeneous. How did the authors quantify the lipid levels in this cell population?

10. For organoid imaging, the authors fix and section for immunostaining. Does fixation affect OPTIR imaging?

11. In Figure 4I, are we looking at the whole organoid without sectioning? Does azide-PA penetrate the depths of the organoid with the same efficiency as the surface? Is the decrease in OPTIR signal solely due to the limitation of optical depth or azide-PA level also contribute? Also, the depths in Figure 4J and 4K are not labeled. Do they correspond to the regions numbered 1-5?

Minor comments:

1. The title "Sub-cellular mapping of lipid metabolites" does not really reflect the content of the results in this manuscript. Sub-cellular mapping would be to show newly synthesized lipids among different parts of the cells, for example, plasma membrane and lipid droplets. The authors should revise their title.

2. In the introduction, the authors mention the importance of human cells and organoids, however, they should still emphasize the significance of traditional animal models, including mice, worms, and flies, in metabolic research. They are especially useful for studying tissue-tissue communication which human cell lines and organoids lack and which carries high importance in metabolic heterogeneity.

3. In the results section, when the authors demonstrate their results of PGRN-knockdown cells, it would be nice to have a 1-2 sentence introduction about PGRN for the audience that may not know.

Reviewer #3 (Remarks to the Author)

Bai and colleagues present an interesting manuscript that describes the optimization of a single cell metabolic imaging platform for high specificity metabolic monitoring. Using this new technique, the authors elegantly show that newly formed lipids can be readily detected in a neuroglioma monolayer paradigm. The authors further show that, using the same technique, newly formed lipids can be measured in human iPSC and iPSC derived microglia. The authors further knock down Granulin (GRN) from iPSC resulting into a different lipid metabolic footprint, showing that this technique could be potentially valuable for disease modelling and understanding. The authors proceed to use OPTIR imaging in iPSC derived organoids where they analyze cell specific lipid concentration and conclude that astrocytes have higher lipid metabolism compared to neurons. These results are consistent with what biologically expected. Overall, I find that this is an interesting manuscript that could make a useful contribution to the field. However, I feel this work could only be suitable for publication after revisions and clarifications of data.

Major points

1) The authors claim to be working with microglia but there is no evidence that the imaged cells are indeed microglia. I understand the authors are using a published protocol, but a lot can go wrong during iPSC differentiation. I would, therefore, advise to add at the very least some immunocytochemistry or, even better, quantitative analysis of canonical microglia markers expression (i.e. IBA1, P2RY12 etc.). There is some good characterization in the Douvaras et al., 2017 stem cell reports that was cited in the methods section.

2) Likewise, there is no evidence that the PGRN-KD iPSCs have lower expression of PGRN compared to PGRN+/+ iPSCs. Please provide qPCR for GRN expression or western blotting for PGRN expression.

3) Bright field images of whole organoids with scale bars would help understanding the size of

these organoids at the time of harvest. The authors conclude that there is progressively lower lipid synthesis towards the core of the organoid. However, I wonder if it would be possible that more than a lipid production issue this is an Azide palmitic penetration issue. Could the authors address if this is among their concerns and if not, why?

4) In figure 4, the authors show single cell analysis within the organoid of astrocytes vs neurons lipid content. However, the analysis would be more convincing if it could be done at a higher magnification. Furthermore, it is particularly hard to pinpoint single neurons with TUJ1 staining, the use of MAP2 or a less diffuse staining is advised.

5) The biological findings in this manuscript confirm the validity of single cell OPTIR imaging as they show what is expected in both PGRN KD and neuron vs astrocytes. However, there have been numerous suggestions that GRN is involved in a few CNS specific mechanisms particularly it has been found that mutations in GRN cause impairment in both astrocytes and microglia (ref: PMC10014110, PMC10014110, PMC10014098 and more), damaging neurons activity. It would add value to this paper to measure lipid production in both organoid and microglia derived from PGRN KD iPSCs. Although labor intensive, this has the potential to uncover important disease phenotypes.

Minor points

- 1) N numbers in the figures is not always clearly specified.
- 2) There is very little detail on statistics.

RESPONSE TO REVIEWERS' COMMENTS

Reviewer #1 (Remarks to the Author)

In their manuscript "Sub-cellular mapping of lipid metabolites using IR probe in human-derived model systems" Yeran Bai et al., report the application of Optical Photothermal Infrared (OPTIR) microscopy in combination with a mid-IR tag, Azide. The application of Azide as vibrational probe facilitates chemical imaging of lipid-fate by improving absorption cross-section over other vibrational probes, such as glucose-d7, by achieving spectral clearance in the spectral silent region of biomolecules. Although the imaging method has been broadly exposed previously and the concept of mid-IR Tags in the silent region has been explored and reported before, I found the combination of OPTIR with Azide a quite promising and timely approach in order to further enhance sensitivity in optothermal imaging (or mid-IR imaging in general) and to expand the library of mid-IR probes. I would like to congratulate the authors on this innovative approach. Overall, the manuscript is well-written, easy to follow and the results are presented with clarity in support of the conclusions. I would like to endorse its publication in Nature Communications as it could be of interest to its broad audience particularly for biotechnology and microscopy communities.

Nevertheless, there are a couple of remarks that might help the authors to improve the work further.

Re: We greatly appreciate the reviewer's recognition of our work and the constructive suggestions. We have addressed the concerns and added more discussions based on the reviewer's comments in the revised manuscript.

1-I would be very curious to see the limit of detection of OPTIR for Azide, as well as the limit of the detection for the tagged lipids, PA for instance. They report adding 100 μ M azide-PA in their neuron/astrocytes experiments, what is the final concentration after adding to the media? What was the azide-TAG concentration for the other experiments? and, more importantly, how do these concentrations compare with the limit of detection? how many times above or below?

Re: We appreciate the reviewer for the inquiry regarding the limit of detection (LoD). We have performed additional experiments and determined the LoD for azide bond of azide-PA in dimethyl sulfoxide (DMSO) is around 100 μ M. We expect better LoD through system optimization in future studies.

The current OPTIR setup is optimized to detect dried sample, where the counter-propagation and epi-detection provides best performance in terms of signal level and spatial resolution. The objectives installed are all air-immersion objective without cover glass correction. In this LoD experiment, we sandwiched serial diluted azide-PA in DMSO between two substrates (top: glass coverslip with thickness of 0.17 mm; bottom: CaF₂ substrate with thickness of 1 mm). Counter-propagation (IR focused from below and visible focused from top) was used. Since in liquid measurement, majority of scattered light travels forward, we used transmission detection where the light is collected with the IR objective.

- Since the visible objective is non-cover glass corrected, the 0.17 mm glass coverslip between the objective (NA 0.8) and sample introduced a distorted visible focus.
- The 1 mm CaF₂ plate between sample and IR reflective objective (NA 0.78) greatly reduced the effective NA of the IR objective and thus reduced the collection efficiency of the transmitted visible light.

- The IR objective serves a dual purpose: focus the IR light, and collect transmitted visible light. The 1mm-thick CaF₂ introduced different focal length shift for IR light and visible light, which made the optimized focus position to be different for the two purposes.

Therefore, in the future LoD and live-cell experiments, we will optimize these factors. One geometry may help to is co-propagation of IR and visible light, and transmission detection with a high NA liquid immersion objective with cover glass correction¹. Prior research has detected 5 μM nitrile in DMSO using an OPTIR setup with a 1.2 NA water-immersion objective and optimal signal detection geometry². We will work on improve the LoD by optimizing sample preparation and system configurations. We have added a Supplementary Figure describing the LoD and added comments in the main manuscript. The supplementary figure and revised manuscript are listed below for reviewer's reference:

Figure S2. Determination of azide-PA detection limit. Azide-PA powder was dissolved in dimethyl sulfoxide (DMSO) at serial dilutions of 2.5 mM, 1 mM, 500 μM , 200 μM , and 100 μM . The solution was sandwiched between two substrates (top: 0.17 mm-thick glass coverslip, bottom: 1 mm-thick CaF₂ plate). Counter propagation (visible focused from top and IR focused from bottom) and transmission detection were used. Spectra were acquired in the range of 2040 to 2300 cm^{-1} . Pure DMSO and azide-PA spectra were acquired to unmix the contribution from azide-PA and DMSO. Linear unmixing was performed to extract the coefficient of azide-PA. At least three locations were acquired for each concentration, and the mean and standard deviation of the azide-PA fitting coefficient are plotted in the curve. At 100 μM , the signal-to-noise ratio (calculated by dividing the mean by standard deviation) is 2.5. The mean value was linear fitted and plotted in red curve.

...We further evaluated the limit of detection (LoD) of azide-PA using spectra from its serial dilution in dimethyl sulfoxide (DMSO) (**Fig. S2**). The LoD of azide bond in DMSO is around 100 μM , which surpasses the LoD of the alkyne bond in SRS microscopy³. The LoD is limited by the relatively low NA objective and non-optimized detection geometry. A recent study has achieved a 5 μM LoD for the nitrile bond in OPTIR microscopy using a 1.2NA water-immersion objective and a fast digitization method². Given the higher extinction coefficient of azide compared to nitrile^{4,5}, we expect an improved LoD with optimized experimental conditions. ' (Page 4, Line 14)

We thank the reviewer for pointing out the potential confusion of the concentration of azide-PA incubation. 100 μM is the final concentration in the media for all the experiments performed. This final concentration is about the same as the limit of detection for the current OPTIR system. We have added this information to the revised manuscript.

2-It not clear how many cells were arbitrary selected for their single-cell spectral analysis; this information might be key to support statistical relevance (with biologists used to measure 10's of thousand to millions of cells).

Re: We thank the reviewer for highlighting this aspect. In the initial manuscript, each dot in the box plot represents an averaged spectrum from a single cell. For each cell, we acquired spectra from least 3 locations, and used these to generate an average spectrum. This spectrum was subsequently utilized for spectral-fitting and area under curve evaluations. To clarity, we've elaborated on this process in the Methods section and have explicitly stated the number of cells utilized for each experiment in figure captions. We recognize the reviewer's emphasis on the importance of cell count in statistical validity. To boost the statistical robustness of our OPTIR application in biomedical contexts, enhancing our setup's throughput is crucial. In the discussion section, we've outlined potential ways for improving imaging speed and provided further discussion on statistical relevance. Please find the revised manuscript below:

'...To derive a spectrum from a single cell, we collected and averaged at least 3 pinpoint spectra after normalization. This resultant spectrum is used for spectral fitting and area under curve calculations.' (Page 18, Line 22)

'To fully explore the potential of OPTIR metabolic imaging platforms for biomedical discovery, we aim to improve the performance of the current system. To provide statistical robustness, the capability to acquire data from a large number of cells in a reasonable time frame is essential. In the present work, our analysis based on tens of cells, was sufficient to suggest a statistical difference between experimental groups. However, more cells may be needed to reveal more subtle biological differences or reveal heterogeneity. For example, cellular experiments that highlight the heterogeneity typically use 10^3 to 10^5 cells^{6,7}...' (Page 12, Line 11)

3-Since focused pump-probe beams were used, this might signify elevated optical/thermal effects to the cells. Have the authors studied the effects of tight irradiation on the cells?

Re: We appreciate the reviewer's attention to this critical detail. We've evaluated the temperature increase on cells based on modulation depth. The calculation suggests a transient temperature of a few Kelvins in the sample. This is consistent with established OPTIR reports and within the safety limit for biological cells. We have added the evaluation and comments in the revised manuscript:

'...Another note on live-cell imaging is the localized temperature increase may raise concerns about interfering with normal cellular functions. We can estimate the thermal effect based on the modulation depth of the OPTIR process. Take a cell image as an example, the DC signal is around 3.5V, and the AC signal is 20 mV, with the AC gain of 2 times, we can calculate the modulation depth to be 2.9×10^{-3} . Since the AC signal is generated based on scattering intensities difference between IR-on and IR-off states, and studies have established the dependence of scattering intensity on temperature to be $10^{-3}/K^{8,9}$, we can estimate our localized temperature increase is around 2.9 K. Additionally, this temperature increase is transient and lasts shorter than the IR pulse width of shorter than 1 μ s. The calculated temperature increase scale is consistent with other photothermal IR reports where different readouts such as fluorescence intensity fluctuation (2-4 K)¹⁰ and quantitate phase (0.1 K)¹¹. Therefore, with the current configuration, we do not expect the thermal effect to significantly impact cellular normal physiology. However, if it is a concern, the local temperature can be reduced by reducing the IR power or shortening the IR pulse width' (Page 13, Line 16)

4-The text is not explicit on whether the cells were alive or fixed; although, by looking at the methods, and considering the imaging speed, it becomes clear that the cells/tissues were fixed. I would appreciate if the authors could clearly state in the script whether the cells are fixed or alive, and (to avoid ambiguity) discuss how it could be applied in live-cell microscopy if not. This is relevant as the use of Azide and OPTIR might become of interest for biologists in longitudinal studies.

Re: We apologize for this oversight and have clearly noted the cell conditions during imaging in the revised manuscript. To clarify, the cells and tissues used in our experiments were indeed fixed for consistency and stability of our samples during data acquisition. We have performed additional experiments on live cells. The cells were HuH7 hepatocellular carcinoma cells and treated with both ^{13}C -glucose (for 48 h) and azide-PA (for 21 h). The cells were imaged without fixation. A representative image is shown in **Fig.R1**. The spectrum from high lipid content showed clear azide signal (2096 cm^{-1}) and lipid ester signal (1744 cm^{-1}), and the peak location was consistent with fixed cells. There's an additional peak around 1700 cm^{-1} compared with the results presented in the main manuscript, which corresponds to the $^{13}\text{C}=\text{O}$ stretching mode that results from ^{13}C -glucose treatment. The lipid contents probing at 1700 cm^{-1} generated from glucose are resulted from *de novo* lipogenesis, and the 44 cm^{-1} peak red shift is due to isotopic effect of ^{13}C and has been reported previously¹². We have added discussion in the revised manuscript on the potential of applying the reported platform to study live-cell metabolism. Sample courtesy: Dr. Caitlin Davis and Sydney Shuster from Yale University.

Figure R1. OPTIR live-cell lipid metabolic imaging. (A-C) Brightfield and OPTIR images at indicated wavenumbers. The region of the OPTIR images acquired is denoted with a red dashed square in (A). (D) Pinpoint OPTIR spectrum from white arrow indicated location in (C).

While the presented work utilizes fixed cells and tissue as a testing bed to demonstrate the feasibility, the platform can be readily translated to study live-cell lipid metabolism. Reports utilizing isotopes to study lipid metabolism and glucose metabolism in live cells have been achieved with OPTIR setups¹²⁻¹⁴. By comparing the spectra and ratios between ^{13}C and ^{12}C ester carbonyls of live cells and fixed cells, Shuster et al. concluded the fixed cells can provide a reliable representation of *de novo* lipogenesis as observed in live cells¹². Spadea et al. have reported the OPTIR spectral comparison between live and fixed cancer cell lines and observed similar protein

amide I to amide II ratios in live and fixed cells¹⁵. These findings suggest that fixed cells can be a good representative of live cells when studying lipid metabolism. However, the fixation process can modify the cell morphology and cause cell shrinkage¹⁶, thus introducing unwanted spatial overlap of targeted molecules and other interfering molecules within the diffraction-limited spots¹². Therefore, to better preserve cell morphology and enable longitudinal studies of lipid metabolism, live-cell imaging will be a preferred option. Additionally, by implementing the throughput-optimizing methods described above, potential motion artifacts that could complicate quantifications in live-cell imaging can be minimized. ...' (Page 13, Line 3)

Reviewer #2 (Remarks to the Author)

This manuscript titled “Sub-cellular mapping of lipid metabolites using IR probe in human-derived model systems” by Yeran Bai and colleagues demonstrates the use of recently developed imaging technology optical photothermal infrared (OPTIR) microscope in detecting and analyzing metabolic heterogeneity, specifically in lipid metabolism, among different cell types. First, the authors revealed that the OPTIR microscope coupled with supplementation of azide-tagged fatty acid (azide-PA) is capable of detecting newly synthesized lipids in an immortalized cell line, specifically human neuroglioma H4 cells. Next, they examined lipid metabolism in hiPSCs and hiPSC-derived microglia and showed that compared to hiPSCs differentiated microglia have significantly higher total lipid and newly synthesized lipid levels. In parallel, they also found that progranulin (PGRN) knockdown in hiPSCs leads to an increase in newly synthesized lipids without altering the level of total lipids, thus to a faster lipid turnover rate. Finally, they utilized the OPTIR microscope to investigate the lipid metabolism in the hiPSC-derived brain organoid model system and showed that lipid droplets containing newly synthesized lipids are found in astrocytes, and not in neurons.

Overall, they demonstrate the use of cutting-edge imaging technology, OPTIR microscope, with several different human-relevant model systems. There is a vast potential in the use of OPTIR metabolic imaging and this manuscript demonstrates some of them, specifically focusing on lipid metabolism. Although they do not reveal a novel biological finding, they open up new avenues for further investigation of lipid metabolism in different cell types, as well as the metabolism of distinct metabolites using different biorthogonal IR tags to assess the metabolic heterogeneity among cells and tissues.

Some of their experimental results need major and minor revisions which are listed below.

Re: We greatly appreciate the reviewer’s very constructive comments and the recognition of the potential of our OPTIR metabolic imaging platform. The feedback has helped us to improve the quality and clarity of our work. We have carefully addressed the questions and concerns the reviewer raised as detailed below.

Major comments:

1. The authors choose to use the azide tag, which has a peak in the cell-silent region, which is great. Does this tag affect the fatty acids to be incorporated into phospholipids? We do not see many signals coming from the plasma membrane. Using OPTIR imaging, is it possible to image membrane incorporation of lipids?

Re: We thank the reviewer’s insightful comments. Yes, we are able to image azide incorporation into the plasma membrane. However, due to relatively lower lipid concentration in plasma

membrane compared to cytoplasmic lipid aggregates such as lipid droplets, the corresponding signal contrast from the membrane is less pronounced.

We have revisited our data, performed a separate contrast adjustment for regions with high and low lipid content, and included a supplementary figure (Fig. S4) to illustrate the successful incorporation of azide-tagged fatty acids into the plasma membrane. When the contrast scale is adjusted to visualize the high lipid regions (0-35 mV), only the high lipid contents such as lipid droplets were visible (Fig. S4A). We then adjusted the contrast to 0-0.5 mV to show the incorporation of azide into low lipid contents regions (Fig. S4B-C). The cellular boundaries were clearly visualized in OPTIR 2096 cm^{-1} channel, and the azide peak is unambiguously resolved when we pin on the cell boundary. As a control, we pin to a cell-free region and observed no peaks in the 2000-2200 cm^{-1} region. The added supplementary figure and the modified main manuscript text is attached below for the reviewer's reference:

Figure S4. Imaging of azide incorporation into plasma membranes via OPTIR.

'...In addition to the incorporation of azide-tagged lipids into intracellular lipid droplets, we also observed their incorporation into plasma membranes (Fig. S4). Since the concentration of lipids is lower in the cell membrane when compared to lipid droplets, the signal contrast is weaker. Despite this, we still managed to observe clear cellular boundaries. Additionally, the azide peak at the cell boundary was unambiguously resolved compared to the no-cell region' (Page 5, Line 4)

2. Figure 1D shows the signal coming from newly synthesized lipids from a single cell. A larger image would have been better. It is not possible to see individual lipid droplets. Also, what type of cell did they use for this experiment? Did the authors co-stain the cells with BODIPY or any other lipid droplet marker to show that the signal is coming from lipid droplets?

Re: We appreciate this comment and have performed additional experiments to validate that the OPTIR lipid signal overlaps with the BODIPY signal. By comparing total lipid (1744 cm^{-1}), newly-synthesized lipid (2096 cm^{-1}), and BODIPY contrast, we confirmed the OPTIR lipid contrasts were indeed from lipids. We have summarized the validation in **Fig. 1D** and added relevant description in the main manuscript:

Figure 1D. Total lipids (1744 cm^{-1}), newly-synthesized lipid (2096 cm^{-1}), and BODIPY staining from a single cell. Scale bars, $10\text{ }\mu\text{m}$.

‘...We first verified that OPTIR image contrasts at 1744 cm^{-1} and 2096 cm^{-1} were indeed from lipids by comparing them with fluorescence images of lipid staining with BODIPY (**Fig. 1D**). A good agreement between OPTIR and BODIPY images validates that the contrasts were from lipids, further underscoring the chemical selectivity of OPTIR imaging...’ (Page 4, Line 31)

3. In Figure 1E, there is a clear intensity difference in OPTIR signal at 1744 cm^{-1} between azide-PA incubated and PA incubated cells. If this signal is coming from total lipids, how did the authors explain the increase in total lipids with azide-PA incubation? A change in total lipids with azide-PA feeding could be misleading for future analyses.

Re: We appreciate the reviewer for this critical observation. For our spectral or image-based quantifications, we did not utilize a single peak or single-color imaging. Since the single-color image does not fully tell the total lipids contents, and will be influenced by biomass and focus difference between fields of views, we always utilized the concept of ‘normalized total lipids’ for quantification, where we divided the 1744 cm^{-1} signal by total protein signal at 1650 cm^{-1} . Based on the normalized lipid ($1740/1650$) quantification (**Fig.1H**), we observed a non-significant difference between azide-PA and PA incubated group. Therefore, we concluded that the total lipid contents will not be influenced by the azide-PA incubation. We have added more explanation on the normalization part in the main manuscript and changed to another representative field of view (**Fig.1F**) to avoid potential ambiguity influenced by the impression of single-color images.

Figure 1F. Representative OPTIR images at 1744 cm^{-1} and 2096 cm^{-1} of human neuroglioma H4 cells after incubated in Azide-PA and PA containing media for 24 h. Corresponding brightfield images are also shown. Scale bars, 20 μm .

‘...To account for focus and biomass variations across different fields of view (FOV), we did not rely solely on a single-color imaging or an individual fitted peak for total lipid quantification. Instead, we normalized the total lipid to protein (1740/1650), thereby producing a more reliable normalized total lipid signal...’ (Page 5, Line 23)

4. This question follows up on the previous one. The concentration of azide-PA is 100 μM . Too much PA is known to be toxic to cells. Did the authors try different concentrations? Maybe lower azide-PA concentrations which do not cause an increase in total lipids should be examined.

Re: We appreciate the reviewer’s concern regarding the concentrations of azide-PA used in our study. We choose a final concentration of 100 μM based on previous vibrational metabolic imaging studies and existing cell viability tests against different PA concentrations. Below are few references to support our decision:

¹⁷ Shi et al. (2020) used 100 μM azide-PA in their cell culture for probing metabolism with a mid-IR microscope.

¹⁸ Stiebing et al. (2017) used 400 μM d_{31} -palmitic acid in their cell culture to image lipid uptake with a stimulated Raman scattering microscope.

¹⁹ Yuan et al. (2021) used stimulated Raman scattering to assess free fatty acid induced lipotoxicity and observed no measurable changes in the survival rate for cells treated with 200 μM PA compared with the untreated group.

²⁰ González-Giraldo et al. (2018) observed a non-significant difference in survival rate for cells treated with 125 μM PA when compared with untreated cells.

While these studies provide evidence of cell viability at 100 μM PA, we acknowledge that PA-induced cellular toxicity can be a concern and the concentration threshold may vary for different cell types. In future investigations for specific biological questions, to address this concern, we will perform cell viability test with different azide-PA concentrations to avoid the potential toxicity.

To explore the potential of using lower concentration azide-PA, we performed additional experiments and confirmed the azide signal is still detectable with 20 μM (final concentration) azide-PA incubation. We have summarized these results in **Fig. S3**, and commented the concentration concern in the main manuscript. The modifications are listed below for the reviewer's reference:

Figure S3. OPTIR imaging with lower azide-PA incubation concentration.

'...This concentration was selected based on prior metabolic studies using vibrational probes and cell viability tests against different PA concentrations¹⁷⁻²⁰. We also observed the incorporation of azide into intracellular lipids at a reduced azide-PA concentration of 20 μM (**Fig. S3**), suggesting that a lower azide-PA incubation concentration is viable for tracing lipid metabolites, especially if there are metabolite-induced viability concerns for certain cell types...' (Page 4, Line 27)

5. For Triacsin C treatment, the authors only included spectral-based quantification. It would be better to include some representative OPTIR images.

Re: We thank the reviewer for the suggestion. We have added representative OPTIR images in **Fig. S6**.

Figure S6. Triacsin C treatment significantly reduces the normalized total lipid and newly-synthesized to total lipids ratio.

‘...We acquired images from both groups and the representative images are shown in **Fig. S6**. It was clear that with the Triacsin C treated group, both the 2096 cm^{-1} and 1744 cm^{-1} contrast drops substantially...’ (Page 6, Line 6)

6. For the time-dependent incorporation of the azide experiment (Figure 2), did the authors use the neuroglioma cells or a different type of cell? Again, why did they observe an increase in total lipids, along with azide-PA levels which indicate newly synthesized lipids? Also, why did the total lipid and newly synthesized lipid levels decrease after 16 hours, if there is a continuous supply of azide-PA in the culture? By 24 hours, both signals were very low. Total lipid levels should not be affected by azide-PA incorporation and if azide-PA is in the culture for the whole 24 hours, the cells should continue to synthesize new lipids.

Re: We thank the reviewer for the insightful questions. For the time-dependent experiment, we used neuroglioma H4 cells. The observed increase in total lipids, along with azide signal, suggests an active synthesis during the initial hours. Our cell culture medium is DMEM supplemented with 10% fetal bovine serum and 100 μM (final concentration) azide-PA. Given that the fatty acid concentration in serum is relatively low (0.1 to 1 μM^{21}) compared to the added azide-PA, the cells are likely uptake more azide-PA over fatty acids present in the serum. This results in the increased newly-synthesized lipid contrast and a higher ratio of newly-synthesized lipid to total lipids for initial treatment hours. The observed decrease in lipids beyond 11 to 16 h could be a consequence of cellular metabolic adjustments in response to limited nutrient availability, but in-depth research is needed for a definitive understanding. We have added more discussion following the results in the revised main manuscript:

'...The observed rise in total lipids and newly-synthesized lipids during the initial hours indicates active lipid synthesis. Since the fatty acid concentration in the cell culture media²¹ is relatively low compared to the added azide-PA, cells are more likely to uptake azide-PA, leading to an increased azide signal. The observed decline of total lipids and newly-synthesized lipids after 11 to 16 h might be attributed to cellular metabolic adaptations due to decreasing nutrient availability. It has been shown that PA treatment can promote cancer cell growth²². With active cellular growth and division, there is an increased demand for lipids, which are essential for membrane expansion and phospholipids synthesis. As exogenous lipid resources are reduced, cells may initiate lipid catabolism to support the increasing lipid demand. This pattern of an initial lipid surge followed by a decline after PA treatment has been described previously in a different cell line²³. Interestingly, the same research highlighted distinct lipid dynamics upon oleic acids (OA) treatment, where it shows a continuous buildup of lipids over 24 h incubation period. These results indicate that specific fatty acids may trigger distinct lipid metabolic pathways, leading to varied lipid dynamics. To thoroughly understand the unique lipid metabolism dynamics that we observed in the present study, it is essential to perform comprehensive analysis across diverse fatty acids and cell types.'

(Page 6, Line 22)

7. Figure 3A shows the representative OPTIR images of newly synthesized lipids in hiPSC and hiPSC-derived microglia. It would be better to have bigger images. It is a little surprising to see such a high level of total lipids and that many lipid droplets in microglia. Can the authors comment on this? They mention about them in the main text as "lipid droplet-like structures". Did they co-stain the cells with any lipid droplet marker or other organelle markers to visualize where OPTIR signals are coming from? Interestingly, not all the "lipid droplet-like structures" contain newly synthesized lipids, as seen in the 2096 cm⁻¹ image. And the ones that have newly synthesized lipids are localized more in the periphery, rather than perinuclear. Is this the case for this specific image or for all?

Re: we appreciate the reviewer's observation and detailed feedback.

- High-level of lipids in microglia: In microglia, the presence of increased lipids has been previously reported²⁴. Additionally, previous studies have suggested the *in vitro* microglia are more activated than their *in vivo* counterparts²⁵, and the activated microglia resulted in increased production of lipid droplets²⁶. This is further validated by our confocal imaging results (**Fig. S9**), showing substantial lipid contents in microglia even without any exogenous fatty acid supplement. Consequently, when external fatty acids, such as azide-PA in our study, are introduced to the culture medium, more pronounced lipid levels in microglia are anticipated.

Figure S9. Representative confocal fluorescence images of hiPSC-derived microglia. The cells were not treated with additional fatty acids. Blue for DAPI and green for BODIPY.

- Lipid-droplet structures: We initially referred to these structures as ‘lipid droplet-like’ based on their signal level. Given that intracellular lipid droplets typically contain a higher concentration of lipids compared to other organelle such as membranes, elevated signals are often indicative of lipid droplets (see Fig. 1D for example). Additionally, these structures produce signals both at 2096 cm^{-1} and 1744 cm^{-1} , signature lipid peaks in IR. However, due to the lack of co-staining with lipid droplet markers in these experiments, we cannot definitively attribute these signals solely to lipid droplets. For clarity, we have removed the ‘lipid droplet-like structure’ in the revised manuscript. In future studies, we plan to incorporate co-staining of lipid droplet markers to provide a more solid understanding of the newly-synthesized lipids across cellular organelles.

‘...In the hiPSC-derived microglia, both the carbonyl and azide signals are prominent, and there is a substantial overlap between the two contrasts (**Fig. S8**). On the other hand, hiPSCs produced a notably reduced signal in both channels....’ (Page 7, Line 10)

- Discrepancy between total lipid and newly-synthesized image: The primary reason for the perceived discrepancy is attributed to different contrast scale. We have added a supplementary figure to illustrate the newly-synthesized lipids largely overlap with the total lipids contrasts. Areas of non-overlap may arise from focus drift during experiments. Moving forward, we will ensure consistent focus across different wavenumbers for the same field of view to minimize focus-drift induced artifacts.

Figure S8. Representative merged OPTIR (green for newly-synthesized lipids and red for total lipids) and brightfield images.

- Lipid distribution: Upon revisiting our data, we observed varied distribution patterns of newly-synthesized lipids, with select examples provided in **Fig. S8**. The intracellular distribution of LipidTOX staining in untreated microglia similarly displayed diverse patterns, as shown in **Fig. S9**. Consequently, we have removed our earlier statements regarding peripheral lipid localization to address this oversight.

8. The finding that total lipids in PGRN-KD cells do not change, but they have a higher lipid turnover rate indicated by increased levels of newly synthesized lipids, is very exciting. If those cells have faster synthesis, they should also have faster lipid catabolism. Is this known from transcriptomic studies? Can the authors elaborate on this finding more in the main text?

Re: We thank the reviewer for the insightful question and suggestion. RNA-sequencing data²⁴ reveals that lipid droplet-enriched microglia in *GRN*-deficient mouse brains show an upregulation in fatty acid degradation pathways, which are closely related to lipid catabolism. These findings offer valuable insights for interpreting our results. However, these studies are based on *GRN*-deficient mouse models. To the best of our knowledge, comprehensive transcriptomic studies on *GRN*-KD hiPSCs or hiPSC-differentiated microglia remains limited. Moving forward, we aim to explore these areas further. We have revised our manuscript accordingly to reflect these findings and their implications:

'Previous studies have indicated high transcriptional similarities between lipid droplet-enriched microglia in *Grn*^{-/-} mouse brains and lipid droplet-accumulating microglia observed in aged mice²⁴. RNA-sequencing (RNA-seq) of *Grn*^{-/-} mice's lipid droplet-enriched microglia revealed significant upregulation of fatty acid degradation-specific genes, suggesting fortified lipid catabolism in these models. Another lipidomic study revealed that *GRN* loss leads to an accumulation of polyunsaturated triacylglycerides, as well as a reduction of diacylglycerides and phosphatidylserines in *GRN* mutant mouse embryonic fibroblasts²⁷. This study also performed RNA-seq and identified a panel of lysosomal genes and lipid metabolic genes that are significantly

dysregulated in *Grn*^{-/-} mouse brains compared to control *Grn*^{+/-} mouse brains. Further, RNA-seq data from age-dependent microglia in *GRN*^{-/-} mice indicated significant upregulation of lysosomal functions (*Ctsb*) and lipid transport (*ApoE*) genes²⁸. These findings indicate an intricate relationship between lipid synthesis, accumulation, breakdown, transport, and *GRN* deficiency. By conducting further transcriptomic analyses on human-relevant *GRN* deficiency models employed in the present study, we can directly correlate our phenotypical OPTIR results with genotypic changes. This approach will expand our understanding of the impact of *GRN* deficiency on lipid metabolism and its potential link to neurodegeneration.' (Page 9, Line 14)

9. Following up on the previous comment, in Figure S4, the resolution of the confocal image of cells is very low. Also, it seems the LipidTox staining was very heterogeneous. How did the authors quantify the lipid levels in this cell population?

Re: We appreciate the reviewer for the critique. The confocal images were taken with step size of 280 nm. The whole field of view is 290 μ m by 290 μ m, and each cell is around 15 μ m in diameter. Additionally, based on our findings in Fig. 3A-C, hiPSC does not accumulate as many as lipids as differentiate cell lines. Therefore, the contrasts of lipids are not as pronounced. We have added two smaller fields of view from the staining results, where each cell is shown more clearly, as well as the lipid distribution in the cells. The quantification method is further clarified in the caption of Fig. S12.

Figure S12 B. Zoom-in views to better show lipid distribution inside cells.

'...(C) Quantification of total lipids. Thresholding masks of lipid contrasts were first created to remove the background signal, and the threshold (pixel value ≥ 30) was consistent for different FOVs as well as different cell lines. The raw lipid images were then multiplied by the corresponding masks and the total intensity within an FOV was integrated. To eliminate the influence of different cell counts on the quantification, we normalized the integrated lipid intensity with the nuclei counts (obtained from DAPI images) in the same FOV.' (Supplementary Information Page 13)

10. For organoid imaging, the authors fix and section for immunostaining. Does fixation affect OPTIR imaging?

Re: We appreciate the reviewer for bring this very important question. We have addressed this concern in our response to Reviewer 1, Point 4. We respectfully direct the reviewer to that section for a comprehensive response. Briefly, we have performed additional experiments with live cells treated with ^{13}C -glucose and azide-PA (**Fig.R1**). Imaging and spectral measurements of these cells without fixation revealed azide and lipid ester signals consistent with the fixed cells. Previous studies^{12,15} have also suggested the fixed cells can provide reliable data similar to live cells. However, we also acknowledge live-cell imaging is advantageous for preserving cell morphology and longitudinal studies.

11. In Figure 4I, are we looking at the whole organoid without sectioning? Does azide-PA penetrate the depths of the organoid with the same efficiency as the surface? Is the decrease in OPTIR signal solely due to the limitation of optical depth or azide-PA level also contribute? Also, the depths in Figure 4J and 4K are not labeled. Do they correspond to the regions numbered 1-5?

Re: We thank the reviewer for pointing out potential ambiguity. In **Fig. 4I**, it's an organoid slice. Therefore, the decrease in OPTIR azide signal is not originated from the optical penetration. We believe the reduced OPTIR azide signal is the combination of reduced azide-PA concentration towards the core, higher cell density at the surface, and altered cellular metabolism due to hypoxia and limited nutrients towards the organoid core. We have made clear description of the organoid slices in the revised manuscript and added comments to discuss the reduced OPTIR azide signal:

'...We took fluorescence images and OPTIR images at different depths from the surface of an organoid slice...' (Page 10, Line 26)

'Given that brain organoids lack a vascularization system, their interior cells are subjected to hypoxia and necrosis from restricted oxygen and nutrient delivery by surface diffusion^{29,30}. This leads to a denser cell population at the periphery. Consequently, the outer cells are naturally the primary absorbers of azide-PA in the cell culture medium, potentially limiting the compound's diffusion into deeper layers. This establishes an azide-PA concentration gradient, with higher levels near the surface and decreasing concentrations toward the core. Previous research has suggested that the viable region of organoids is typically limited to a few hundred μm from the surface, despite efforts to enhance oxygen and nutrient diffusion³¹⁻³³. Given this context, it is possible that metabolism towards the core is compromised due to hypoxia and nutrient scarcity. Therefore, the observed reduction of OPTIR azide signal towards the core is likely a complex interplay between cell density, azide-PA concentration gradient, and potential metabolic alterations due to oxygen and nutrient deficiency. In future studies, we aim to integrate organoid slices into our metabolic research. These slices can access oxygen and nutrients from both top and bottom surfaces and have been shown to substantially reduce internal cell death³³.' (Page 11, Line 1)

We apologize for the oversight of the labelling in **Fig. 4J-K**. The three field of view (from top to bottom) corresponds to the three dashed region in **Fig. 4I** (from right to left). The number 1-5 in **Fig. 4I** is the spectral location presented in **Fig. 4L**. We have added labels in the figure and descriptions in the figure captions. The revised figure:

Figure 4I-L. Fluorescence and OPTIR images along with spectra data of an organoid slice.

Minor comments:

1. The title “Sub-cellular mapping of lipid metabolites” does not really reflect the content of the results in this manuscript. Sub-cellular mapping would be to show newly synthesized lipids among different parts of the cells, for example, plasma membrane and lipid droplets. The authors should revise their title.

Re: We thank the reviewer for the suggestion. Although we have some preliminary data on the sub-cellular mapping of lipid metabolites, we are not focusing on discovering the azide incorporation into different sub-cellular compartments. We have changed ‘sub-cellular’ to ‘single-cell’ in our title.

2. In the introduction, the authors mention the importance of human cells and organoids, however, they should still emphasize the significance of traditional animal models, including mice, worms, and flies, in metabolic research. They are especially useful for studying tissue-tissue communication which human cell lines and organoids lack and which carries high importance in metabolic heterogeneity.

Re: We agree with the reviewer on the importance of animal research in metabolic research. We have modified our introduction to reflect this very important body of research:

‘...Traditional animal models engineered to produce phenotypic features of human diseases like *Caenorhabditis elegans*, *Drosophila melanogaster*, and mice have been foundational in metabolic research³⁴⁻³⁸. They have been extensively characterized, providing a vast pool of resources and a unique environment to study tissue interactions...’ (Page 1, Line 16)

3. In the results section, when the authors demonstrate their results of PGRN-knockdown cells, it would be nice to have a 1-2 sentence introduction about PGRN for the audience that may not know.

Re: We appreciate the reviewer’s suggestion and have incorporated more introduction to *GRN* in the results section:

'...PGRN, encoded by the *GRN* gene, is widely expressed in various tissues including those in the central nervous systems, where it is predominantly found in neurons and microglia³⁹⁻⁴¹. PGRN plays a vital role in many physiological processes including regulating lysosomal functions and inflammation. Critically, deficiencies in *GRN* have been linked to a range of neurodegenerative diseases including frontotemporal dementia and Alzheimer's disease³⁹⁻⁴¹. *GRN* and PGRN have been closely associated with lipid metabolism. For example, complete loss of PGRN leads to Neuronal Ceroid Lipofuscinosis^{42,43}, a neurodegenerative disease characterized by lysosomal accumulation of lipofuscin, a lipid-protein aggregate. Moreover, previous studies have indicated lipid accumulation in humans and mice with *GRN* deficiency, as evidenced through lipid staining and lipidomic methodologies^{24,27}...' (Page 7, Line 29)

Reviewer #3 (Remarks to the Author)

Bai and colleagues present an interesting manuscript that describes the optimization of a single cell metabolic imaging platform for high specificity metabolic monitoring. Using this new technique, the authors elegantly show that newly formed lipids can be readily detected in a neuroglioma monolayer paradigm. The authors further show that, using the same technique, newly formed lipids can be measured in human iPSC and iPSC derived microglia. The authors further knock down Granulin (*GRN*) from iPSC resulting into a different lipid metabolic footprint, showing that this technique could be potentially valuable for disease modelling and understanding. The authors proceed to use OPTIR imaging in iPSC derived organoids where they analyze cell specific lipid concentration and conclude that astrocytes have higher lipid metabolism compared to neurons. These results are consistent with what biologically expected. Overall, I find that this is an interesting manuscript that could make a useful contribution to the field. However, I feel this work could only be suitable for publication after revisions and clarifications of data.

Re: We sincerely appreciate the reviewer's acknowledgement of our methodology. We also are grateful for the insightful questions and suggestions provided, as addressing them has enhanced the clarity and scientific accuracy of the manuscript.

Major points

1) The authors claim to be working with microglia but there is no evidence that the imaged cells are indeed microglia. I understand the authors are using a published protocol, but a lot can go wrong during iPSC differentiation. I would, therefore, advise to add at the very least some immunocytochemistry or, even better, quantitative analysis of canonical microglia markers expression (i.e. IBA1, P2RY12 etc.). There is some good characterization in the Douvaras et al., 2017 stem cell reports that was cited in the methods section.

Re: We thank the reviewer for the suggestion. We have performed additional immunocytochemistry experiments to assess the expression of canonical microglia markers including IBA1 and CD45. The results confirm the successful differentiation of the hiPSCs into microglia. The results of these characterization results have been incorporated into **Fig. S7**. Since we have performed additional experiments of microglia using another protocol⁴⁴ (related to **Fig. 3I**), we also conducted immunocytochemistry imaging of these cells (**Fig. S13**). The added characterizations are listed below for the reviewer's reference:

Figure S7. Representative confocal fluorescence imaging of hiPSC-derived microglia and hiPSCs immunostained for canonical microglia markers IBA1 and CD45.

Figure S13. Representative confocal fluorescence images of induced-transcription factor microglia-like cells (iTF-MG) and hiPSCs immunostained for canonical microglia markers IBA1 and CD45.

2) Likewise, there is no evidence that the PGRN-KD iPSCs have lower expression of PGRN compared to PGRN+/+ iPSCs. Please provide qPCR for GRN expression or western blotting for PGRN expression.

Re: We acknowledge the reviewer's point regarding the validation. To address this concern, we have performed qPCR assays to evaluate the *GRN* expression. Our results indicate a significant reduction of *GRN* expression in *GRN*-KD hiPSCs (Fig. S11, related to Fig. 3E). Additionally, we also quantified the *GRN* expression in *GRN*-KD microglia cells (Fig. S14, related to Fig. 3I).

Figure S11. qPCR characterization of *GRN* expression levels in Ctrl and *GRN*-KD hiPSCs confirmed successful knockdown of *GRN*.

Figure S14. qPCR characterization of *GRN* expression levels in Ctrl and *GRN*-KD induced-transcription factor (iTF)-microglia confirmed successful knockdown of *GRN*.

3) Bright field images of whole organoids with scale bars would help understanding the size of these organoids at the time of harvest. The authors conclude that there is progressively lower lipid synthesis towards the core of the organoid. However, I wonder if it would be possible that

more than a lipid production issue this is an Azide palmitic penetration issue. Could the authors address if this is among their concerns and if not, why?

Re: We thank the reviewer's suggestion. The organoids we used in the manuscript were 3-4 mm in diameter on average. We did not take a photo prior to harvesting, but we have photos from other batches following the same protocol as described in the manuscript. We have incorporated a photo of these organoids in **Fig. S15**. The organoids were imaged at similar age at the organoids we used in the manuscript (6 months vs. 5.5 months). For future studies, we will make sure to take brightfield images prior to harvest, so that the readers can have a clear understanding of the organoid size and morphology.

'...By 5.5 months, the brain organoid has expanded to an average diameter of 3 to 4 mm (**Fig. S15**) and showed an abundant network of organized neurons and astrocytes ...' (Page 10, Line 4)

Figure S15. Brightfield images of brain organoids.

Regarding the azide-PA penetration, we agreed with the reviewer that it's one of the contributing factors to the decreased OPTIR 2096 cm^{-1} signal. We have addressed this concern in our response to Reviewer 2, Point 11. We respectfully direct the reviewer to that section for a detailed response, or they can refer to Page 11, Line 1 in the main manuscript. Briefly, we have added additional discussion regarding potential reasons for the reduced OPTIR azide signal towards core. We have attributed it to factors including reduced azide-PA concentration towards core, cell population gradient, and potential metabolism alteration due to hypoxia and nutrient scarcity towards core.

4) In figure 4, the authors show single cell analysis within the organoid of astrocytes vs neurons lipid content. However, the analysis would be more convincing if it could be done at a higher magnification. Furthermore, it is particularly hard to pinpoint single neurons with TUJ1 staining, the use of MAP2 or a less diffuse staining is advised.

Re: We appreciate the reviewer for the suggestion. We have performed additional experiments using MAP2 as a neuronal cell marker, and the results are presented in **Fig. S16**. Upon repeating the Pearson correlation analysis with these MAP2 images, we found the results are consistent with our initial findings with TUJ1 staining: the newly-synthesized lipids overlap significantly more with astrocytes than neurons. The revised manuscript and added supplementary figures are listed below for the reviewer's reference:

'...We further validated our findings with another less diffusive neuronal marker MAP2 and found consistent results (**Fig. S16**)...' (Page 10, Line 21)

Figure. S16. Representative field of views for astrocyte (GFAP), neuron (MAP2), newly-synthesized lipids (OPTIR 2096 cm^{-1}), and merged images. Pearson correlation analysis showed significant higher colocalization of newly-synthesized lipids with astrocytes than neurons.

5) The biological findings in this manuscript confirm the validity of single cell OPTIR imaging as they show what is expected in both PGRN KD and neuron vs astrocytes. However, there have been numerous suggestions that GRN is involved in a few CNS specific mechanisms particularly it has been found that mutations in GRN cause impairment in both astrocytes and microglia (ref: PMC10014110, PMC10014110, PMC10014098 and more), damaging neurons activity. It would add value to this paper to measure lipid production in both organoid and microglia derived from PGRN KD iPSCs. Although labor intensive, this has the potential to uncover important disease phenotypes.

Re: We appreciate the reviewer's insights and the suggestion. We have performed additional experiments on microglia cells differentiated from *GRN*-KD hiPSCs, and the results is summarized in **Fig. 3I-K**. Consistent with existing reports, our single-cell OPTIR imaging results showed a significant accumulation of lipids in *GRN*-KD microglia. Notably, we have also identified significant increase in newly-synthesized lipid to total lipid ratio for *GRN*-KD microglia, providing direct evidence of increased lipid metabolism in these cells. The revised manuscript and added data are listed below for the reviewer's reference:

'Building on these insights, we further investigated the implications of *GRN*-KD in a differentiated cell state. Growing evidence has suggested the pivotal role of microglia in the disease pathogenesis of frontotemporal dementia with *GRN* mutations, and studies have indicated that *Grn*^{-/-} microglia accumulate significantly more lipids^{24,28,45,46}. This led us to investigate if the lipid metabolic alterations observed in hiPSCs were also evident in differentiated microglia cells. Adopting an established protocol⁴⁴, we derived microglia from a CRISPRi hiPSC cell line that

allows rapid generation of microglia through inducible expression of transcription factors and allows knockdown of endogenous genes. These induced-transcription factor (iTF) microglia-like cells show ramified morphology and express canonical microglia markers (**Fig. S13**). As shown in **Fig. S14**, *GRN* expression levels were successfully knocked down in *GRN*-KD microglia cells. The *GRN*-KD and control iTF-microglia cells were incubated in azide-PA (final concentration 100 μ M) containing media for 24 hours before being fixed for OPTIR imaging. Representative OPTIR images at 1744 cm^{-1} , 2096 cm^{-1} , 1654 cm^{-1} , and ratioed images are shown in **Fig. 3I**. An increased contrast was evident in both lipid-associated channels for *GRN*-KD cells. The ratioed results clearly illustrate increased levels of normalized total lipids and newly-synthesized lipid to total lipids ratio. We further performed statistical analysis comparing the ratio of newly-synthesized lipids to total lipids, averaged across single-cell areas, between the control and *GRN*-KD groups (**Fig. 3J-K**). Consistent with both imaging results and previous literature²⁴, the normalized total lipids revealed a significant lipid accumulation in the *GRN*-KD iTF-microglia when compared to control cells (**Fig. 3J**). Moreover, the increased newly-synthesized to total lipids ratio provides direct evidence of the increased lipid metabolism associated with *GRN*-KD microglia cells.’ (Page 8, Line 27)

Figure. 3I-K. OPTIR imaging and quantification of *GRN*-KD and control induced-transcription factor (iTF)-microglia.

Additionally, we also added a paragraph to interpret the *GRN*-KD hiPSC and hiPSC-derived microglia data, and correlates with previous studies. Please refer to Response to Reviewer 2, Point 8, or main manuscript Page 9, Line 14.

Given the prolonged maturation period required for brain organoids, we plan to incorporate this study into our future research. We've included a discussion in the revised manuscript to address this valuable suggestion:

‘... In this study, we identified an elevated lipid metabolism in *GRN*-deficient hiPSCs and hiPSC-derived microglia cells. Moving forward, we aim to explore if similar phenotypes are evident in other cell types such as *GRN*-KD astrocytes, which have been shown to promote synaptic dysfunction^{45,47}. To deepen our insights, we plan to incorporate more complex systems such as hiPSC-derived brain organoids and animal models^{24,27,47}. Through these investigations, we aim to explore how *GRN* deficiency differentially impacts lipid metabolism across cell types, brain regions, and model systems. These insights could potentially shed light on the underlying mechanisms of neurodegenerative diseases linked to *GRN* mutation and offer new avenues for therapeutic interventions ...’ (Page 13, Line 29)

Minot points

1) N numbers in the figures is not always clearly specified.

Re: We apologize for this oversight. We have now clearly specified n numbers for each experiment in all relevant figure captions.

2) There is very little detail on statistics.

Re: We apologize for this oversight. We have now revised the figure captions to provide specific details about box plot data representation and the statistical analysis methods utilized.

References:

- 1 Liebel, M., Toninelli, C. & van Hulst, N. F. Room-temperature ultrafast nonlinear spectroscopy of a single molecule. *Nature Photonics* **12**, 45-49 (2018).
- 2 He, H. *et al.* in *Advanced Chemical Microscopy for Life Science and Translational Medicine 2023*. PC123920S (SPIE).
- 3 Wei, L. *et al.* Live-cell imaging of alkyne-tagged small biomolecules by stimulated Raman scattering. *Nature methods* **11**, 410-412 (2014).
- 4 Schmitz, A. J. *et al.* Two-dimensional infrared study of vibrational coupling between azide and nitrile reporters in a RNA nucleoside. *The Journal of Physical Chemistry B* **120**, 9387-9394 (2016).
- 5 Gai, X. S., Coutifaris, B. A., Brewer, S. H. & Fenlon, E. E. A direct comparison of azide and nitrile vibrational probes. *Physical Chemistry Chemical Physics* **13**, 5926-5930 (2011).
- 6 Huang, K.-C., Li, J., Zhang, C., Tan, Y. & Cheng, J.-X. Multiplex stimulated Raman scattering imaging cytometry reveals lipid-rich protrusions in cancer cells under stress condition. *Iscience* **23** (2020).
- 7 Nitta, N. *et al.* Intelligent image-activated cell sorting. *Cell* **175**, 266-276. e213 (2018).
- 8 Yin, J. *et al.* Nanosecond-resolution photothermal dynamic imaging via MHz digitization and match filtering. *Nature Communications* **12**, 1-11 (2021).
- 9 Zong, H. *et al.* Background-Suppressed High-Throughput Mid-Infrared Photothermal Microscopy via Pupil Engineering. *ACS Photonics* **8**, 3323-3336 (2021).
- 10 Zhang, Y. *et al.* Fluorescence-Detected Mid-Infrared Photothermal Microscopy. *Journal of the American Chemical Society* **143**, 11490-11499 (2021).
- 11 Tamamitsu, M. *et al.* Label-free biochemical quantitative phase imaging with mid-infrared photothermal effect. *Optica* **7**, 359-366 (2020).
- 12 Shuster, S. O., Burke, M. J. & Davis, C. M. Spatiotemporal Heterogeneity of De Novo Lipogenesis in Fixed and Living Single Cells. *The Journal of Physical Chemistry B* **127**, 2918-2926 (2023).
- 13 Bai, Y., Zhang, D., Li, C., Liu, C. & Cheng, J.-X. Bond-selective imaging of cells by mid-infrared photothermal microscopy in high wavenumber region. *The Journal of Physical Chemistry B* **121**, 10249-10255 (2017).
- 14 Lima, C., Muhamadali, H., Xu, Y., Kansiz, M. & Goodacre, R. Imaging isotopically labeled bacteria at the single-cell level using high-resolution optical infrared photothermal spectroscopy. *Analytical Chemistry* **93**, 3082-3088 (2021).
- 15 Spadea, A., Denbigh, J., Lawrence, M. J., Kansiz, M. & Gardner, P. Analysis of fixed and live single cells using optical photothermal infrared with concomitant Raman spectroscopy. *Analytical Chemistry* **93**, 3938-3950 (2021).
- 16 Kim, S.-O., Kim, J., Okajima, T. & Cho, N.-J. Mechanical properties of paraformaldehyde-treated individual cells investigated by atomic force microscopy and scanning ion conductance microscopy. *Nano Convergence* **4**, 1-8 (2017).
- 17 Shi, L. *et al.* Mid-infrared metabolic imaging with vibrational probes. *Nature Methods* **17**, 844-851 (2020).
- 18 Stiebing, C. *et al.* Real-time Raman and SRS imaging of living human macrophages reveals cell-to-cell heterogeneity and dynamics of lipid uptake. *Journal of Biophotonics* **10**, 1217-1226 (2017).
- 19 Yuan, Y., Shah, N., Almohaisin, M. I., Saha, S. & Lu, F. Assessing fatty acid-induced lipotoxicity and its therapeutic potential in glioblastoma using stimulated Raman microscopy. *Scientific Reports* **11**, 7422 (2021).
- 20 González-Giraldo, Y., Garcia-Segura, L. M., Echeverria, V. & Barreto, G. E. Tibolone preserves mitochondrial functionality and cell morphology in astrocytic cells treated with palmitic acid. *Molecular Neurobiology* **55**, 4453-4462 (2018).

- 21 Freshney, R. I. *Culture of animal cells: a manual of basic technique and specialized applications*. (John Wiley & Sons, 2015).
- 22 Fatima, S. *et al.* High-fat diet feeding and palmitic acid increase CRC growth in β 2AR-dependent manner. *Cell Death & Disease* **10**, 711 (2019).
- 23 Eynaudi, A. *et al.* Differential effects of oleic and palmitic acids on lipid droplet-mitochondria interaction in the hepatic cell line HepG2. *Frontiers in Nutrition* **8**, 901 (2021).
- 24 Marschallinger, J. *et al.* Lipid-droplet-accumulating microglia represent a dysfunctional and proinflammatory state in the aging brain. *Nature Neuroscience* **23**, 194-208 (2020).
- 25 Maguire, E. *et al.* Assaying microglia functions in vitro. *Cells* **11**, 3414 (2022).
- 26 Ralhan, I., Chang, C.-L., Lippincott-Schwartz, J. & Ioannou, M. S. Lipid droplets in the nervous system. *Journal of Cell Biology* **220**, e202102136 (2021).
- 27 Evers, B. M. *et al.* Lipidomic and transcriptomic basis of lysosomal dysfunction in progranulin deficiency. *Cell Reports* **20**, 2565-2574 (2017).
- 28 Zhang, J. *et al.* Neurotoxic microglia promote TDP-43 proteinopathy in progranulin deficiency. *Nature* **588**, 459-465 (2020).
- 29 Chiaradia, I. & Lancaster, M. A. Brain organoids for the study of human neurobiology at the interface of in vitro and in vivo. *Nature Neuroscience* **23**, 1496-1508 (2020).
- 30 Qian, X., Song, H. & Ming, G.-I. Brain organoids: advances, applications and challenges. *Development* **146**, dev166074 (2019).
- 31 Giandomenico, S. L. *et al.* Cerebral organoids at the air-liquid interface generate diverse nerve tracts with functional output. *Nature Neuroscience* **22**, 669-679 (2019).
- 32 Jacob, F. *et al.* A patient-derived glioblastoma organoid model and biobank recapitulates inter-and intra-tumoral heterogeneity. *Cell* **180**, 188-204. e122 (2020).
- 33 Qian, X. *et al.* Sliced human cortical organoids for modeling distinct cortical layer formation. *Cell Stem Cell* **26**, 766-781. e769 (2020).
- 34 Mutlu, A. S., Duffy, J. & Wang, M. C. Lipid metabolism and lipid signals in aging and longevity. *Developmental Cell* **56**, 1394-1407 (2021).
- 35 Musselman, L. P. & Kühnlein, R. P. Drosophila as a model to study obesity and metabolic disease. *Journal of Experimental Biology* **221**, jeb163881 (2018).
- 36 Ashrafi, K. Obesity and the regulation of fat metabolism. *WormBook: The Online Review of C. elegans Biology [Internet]* (2007).
- 37 Li, Y. *et al.* Direct imaging of lipid metabolic changes in Drosophila ovary during aging using DO-SRS microscopy. *Frontiers in Aging* **2**, 819903 (2022).
- 38 Shi, L. *et al.* Optical imaging of metabolic dynamics in animals. *Nature Communications* **9**, 2995 (2018).
- 39 Rhinn, H., Tatton, N., McCaughey, S., Kurnellas, M. & Rosenthal, A. Progranulin as a therapeutic target in neurodegenerative diseases. *Trends in Pharmacological Sciences* **43**, 641-652 (2022).
- 40 Cenik, B., Sephton, C. F., Cenik, B. K., Herz, J. & Yu, G. Progranulin: a proteolytically processed protein at the crossroads of inflammation and neurodegeneration. *Journal of Biological Chemistry* **287**, 32298-32306 (2012).
- 41 Kao, A. W., McKay, A., Singh, P. P., Brunet, A. & Huang, E. J. Progranulin, lysosomal regulation and neurodegenerative disease. *Nature Reviews Neuroscience* **18**, 325-333 (2017).
- 42 Smith, K. R. *et al.* Strikingly different clinicopathological phenotypes determined by progranulin-mutation dosage. *The American Journal of Human Genetics* **90**, 1102-1107 (2012).
- 43 Almeida, M. R. *et al.* Portuguese family with the co-occurrence of frontotemporal lobar degeneration and neuronal ceroid lipofuscinosis phenotypes due to progranulin gene mutation. *Neurobiology of Aging* **41**, 200. e201-200. e205 (2016).

- 44 Dräger, N. M. *et al.* A CRISPRi/a platform in human iPSC-derived microglia uncovers regulators of disease states. *Nature neuroscience* **25**, 1149-1162 (2022).
- 45 Pinarbasi, E. S. & Barmada, S. J. Glia in FTLD-GRN: from supporting cast to leading role. *The Journal of Clinical Investigation* **133** (2023).
- 46 Lui, H. *et al.* Progranulin deficiency promotes circuit-specific synaptic pruning by microglia via complement activation. *Cell* **165**, 921-935 (2016).
- 47 Marsan, E. *et al.* Astroglial toxicity promotes synaptic degeneration in the thalamocortical circuit in frontotemporal dementia with GRN mutations. *The Journal of Clinical Investigation* **133** (2023).

REVIEWERS' COMMENTS

Reviewer #1 (Remarks to the Author):

The authors have addressed all my comments and attended my suggestions which now re-enforce the quality and clarity of the paper. I have no further comments and in my opinion the paper can be published, however I have one final suggestion:

in Page 4 line 19-20 the authors wrote

"...we expect an improved LoD with optimized experimental conditions"

this is a vague line. If the authors could be more explicitly state what strategies they will or could implement to obtain "optimized experimental conditions", that will improve further their work.

I wish the authors all success.

Reviewer #2 (Remarks to the Author):

The authors have addressed all my comments with clear explanations. With the corrections and improvements in both text and figures, the manuscript can be accepted for publication.

Reviewer #3 (Remarks to the Author):

The authors have fully addressed all my concerns.

RESPONSE TO REVIEWERS' COMMENTS

Reviewer #1 (Remarks to the Author):

The authors have addressed all my comments and attended my suggestions which now re-enforce the quality and clarity of the paper. I have no further comments and in my opinion the paper can be published, however I have one final suggestion:

in Page 4 line 19-20 the authors wrote

"...we expect an improved LoD with optimized experimental conditions" this is a vague line. If the authors could be more explicitly state what strategies they will or could implement to obtain "optimized experimental conditions", that will improve further their work.

I wish the authors all success.

Re: We sincerely appreciate the reviewer's recognition and encouraging words. We are grateful for the reviewer's suggestion to clarify the specific strategies for achieving optical LoD. We have elaborated on this point with additional details:

'...Given the higher extinction coefficient of azide compared to nitrile, we expect an improved LoD with optimized experimental conditions such as co-propagation of IR and visible light, along with transmission detection using a high numerical aperture liquid immersion objective lens with cover glass correction. Such an approach is anticipated to enhance the photon collection efficiency, thereby improving the LoD of target molecules.¹' (Page 4, Line 19)

Reviewer #2 (Remarks to the Author):

The authors have addressed all my comments with clear explanations. With the corrections and improvements in both text and figures, the manuscript can be accepted for publication.

Re: We would like to express our sincere appreciation once again for the reviewer's thorough and constructive feedback on our first draft, which has significantly contributed to the improvement of our manuscript. We are pleased to know that the reviewer finds the revised manuscript suitable for publication.

Reviewer #3 (Remarks to the Author):

The authors have fully addressed all my concerns.

Re: We are grateful for the reviewer's acknowledgment. The reviewer's insightful suggestions have been instrumental in enhancing the quality and clarity of our manuscript.

1 Liebel, M., Toninelli, C. & van Hulst, N. F. Room-temperature ultrafast nonlinear spectroscopy of a single molecule. *Nature Photonics* 12, 45-49 (2018).